# Progressive decomposition of infrared and visible image fusion network with joint transformer and Resnet

Fang Zhu[1], Wei Liu[2]*

1 Department of Mathematics, Ministry of General Education, Anhui Xinhua University, Hefei, China,
2 College of Mathematics and Computer Science, Tongling University, Tongling, China

* lw_feixi@163.com

## Abstract

The objective of image fusion is to synthesize information from multiple source images into a single, high-quality composite that is information-rich, thereby enhancing both human visual interpretation and machine perception capabilities. This process also establishes a robust foundation for downstream image-related tasks. Nevertheless, current deep learning-based networks frequently neglect the distinctive features inherent in source images, presenting challenges in effectively balancing the interplay between basic and detailed features. To tackle this limitation, we introduce a progressive decomposition network that integrates Lite Transformer (LT) and ResNet architecture for infrared and visible image fusion (IVIF). Our methodology unfolds in three principal stages: Initially, a foundational convolutional neural network (CNN) is deployed to extract coarse-scale features from the source images. Subsequently, the LT is employed to bifurcate these coarse features into basic and detailed feature components. In the second phase, to augment the detail information across various inter-layer extractions, we substitute the conventional ResNet preprocessing with a combination of coarse and LT module. Cascade LT operations are implemented following the initial two ResNet blocks (ResB), enabling two-branch feature extraction from these reconfigured blocks. The final stage involves the design of specialized fusion sub-networks to process the basic and detail information blocks extracted from different layers. These processed image feature blocks are then channeled through semantic injection module (SIM) and Transformer decoders to generate the fused image. Complementing this architecture, we have developed a semantic information extraction module that aligns with the progressive inter-layer detail extraction framework. The LT module is strategically embedded within the ResNet network architecture to optimize the extraction of both basic and detailed features across diverse layers. Moreover, we introduce a novel correlation loss function that operates on the basic and detail information between layers, facilitating the correlation of basic features while maintaining the independence of detail features across layers. Through comprehensive qualitative and quantitative analyses conducted on multiple

**Data availability statement:** All relevant data for this study are publicly available from the GitHub repository (https://github.com/I-am-buzzy/Progressive-Decomposition-Network-with-Joint-Transformer-and-Resnet) and the Supporting information files.

**Funding:** The authors would like to extend their heartfelt appreciation to the editorial boardandanonymous reviewers for their meticulous evaluation, valuable insights, andconstructive recommendations, which have significantly enhanced the quality of this work. This research was financially supported by the Natural Science ResearchKeyProgram of the Anhui Provincial Department of Education (Grant Nos. 2024AH050611, 2022AH051750, 2023AH051784 and 2023AH051807), partiallyfunded by the Fundamental Research Funds for Tongling University (No. 2022tlxy-rc11) and Anhui Xinhua University Quality Engineering Project (Grant Nos. 2024jy011 and 2024hhkcx01).

**Competing interests:** The authors have declared that no competing interests exist.

infrared-visible datasets, we demonstrate the superior potential of our proposed network for advanced visual tasks. Our network exhibits remarkable performance in detail extraction, significantly outperforming existing deep learning methodologies in this domain.

## 1. Introduction

Different imaging devices have unique imaging mechanisms, resulting in captured images with distinct characteristics that reflect information from various perspectives. The primary objective of image fusion research has consistently been to extract richer and more comprehensive information from multi-modal images [1–4]. In challenging external environments, many devices are limited to capturing only partial information characteristics of an image based on their inherent capabilities. For example, unmanned equipment [2] equipped with camera technology designed to handle complex field scenes may face difficulties in accurately locating and identifying concealed objects due to uncontrollable factors, thereby presenting significant challenges to scientific research. To address these limitations, researchers have found that infrared imaging can effectively capture prominent target objects based on thermal radiation, offering notable advantages in terms of anti-interference and anti-obscuration. This capability partially mitigates the shortcomings associated with outdoor imaging. However, infrared imaging is often limited in its ability to describe environmental details, frequently resulting in the surroundings of the target object appearing blurred and indistinct. In contrast, visible light imaging excels at capturing rich texture and detail information, providing a complementary perspective that enhances overall image quality and utility. Consequently, numerous researchers have capitalized on the complementary characteristics of these two imaging modalities to devise diverse fusion methodologies. These approaches are designed to enhance target prominence while simultaneously delivering comprehensive contextual information, effectively addressing the constraints inherent in single-modal visual techniques.

The infrared and visible light images in Fig 1 are representative image pairs selected from the classic TNO dataset. Fig 1 illustrates a typical example of IVIF in a complex and harsh field environment. Under such conditions, visible light can only capture the texture information around the fog, making it difficult to perceive the characteristics of the target behind the fog, with much of the detailed information obscured. Infrared images, due to their unique imaging mechanism, can capture an overall view of the target through the fog based on thermal radiation. As demonstrated in Fig 1c-e, image fusion effectively aggregates information from both modal images. Clearly, effective fusion methods can highlight the features of the target while also describing the detailed information of the environment.

Fig 1 provides a visual comparison of different fusion methods. To quantitatively validate the superiority of our approach, we employ two rigorous evaluation metrics: Kullback-Leibler Divergence (KL) for measuring distributional differences between images and Centered Kernel Alignment (CKA) for assessing their structural similarity.

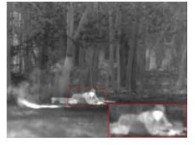 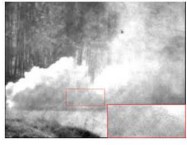 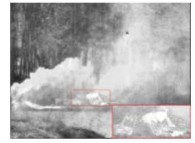 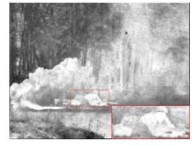 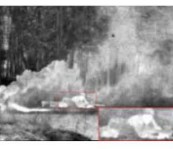

| (a)Infrared | (b)visible | (c)CCDFusion | (d)PSFusion | (e)Ours |

**Fig 1. Provides a visual comparison of different fusion methods.**

As shown in Table 1, which presents the average KL and CKA values between fused and original images, CCDFusion demonstrates significantly higher KL divergence (indicating poorer distribution matching) and lower CKA values (reflecting weaker structural similarity) compared to other methods. Our proposed method, by contrast, achieves optimal performance on both metrics, with the lowest KL divergence and highest CKA values, thereby conclusively demonstrating the effectiveness of our fusion strategy. In this paper, the term "basic features" refers to cross-modal features with CKA>0.6, while "detailed features" denote those with CKA<0.3.

Currently, several advanced deep learning-based IVIF frameworks have demonstrated remarkable capabilities in target enhancement and texture preservation. Notable examples include adversarial mutual constraint-based fusion networks, deep learning architectures emphasizing texture and intensity ratio preservation [5–7], and saliency-guided fusion frameworks [8]. However, these approaches predominantly focus on optimizing fusion quality metrics while overlooking two critical aspects: inter-modal variability and the intrinsic relationship between image fusion and subsequent high-level vision tasks. Such limitations often result in fusion outcomes with substantial detail loss and inadequate target saliency. In response to these challenges, Luo *et al.* [9] proposed a novel multi-branch encoder with contrastive constraints, specifically designed to learn both basic and detailed features across modalities for multi-modal image fusion. While innovative, their approach presents certain limitations: the experimental validation methodology appears outdated and lacks robustness, and the network architecture remains relatively conventional in its monolithic design. A significant advancement was achieved by Zhao *et al.* [10] through their correlation-driven fusion network (CCDFusion), which redefined traditional fusion paradigms by explicitly modeling modality-shared and modality-specific features. Despite its impressive fusion performance, CCDFusion's architectural design shows room for improvement, particularly in its simplistic approach to base and detail encoding where information from different modalities is merely summed and concatenated before decoding. As illustrated in Fig 1, this architecture demonstrates sub-optimal performance in detail extraction compared to its counterparts. In contrast, PSFusion [11] exhibits superior capability in capturing detailed target information, as evidenced by its clearer representation of fog dispersion and enhanced visibility of vegetation in bush areas. This comparative analysis underscores the pivotal role of fusion methodology design in effective information aggregation and highlights the need for more sophisticated architectural approaches in IVIF systems.

Motivated by the aforementioned challenges, we propose a novel and robust framework designed to overcome the limitations of incomplete extraction of basic and detailed features in existing networks. Our primary objective is to enhance texture detail preservation while establishing a robust foundation for downstream image processing tasks. The proposed network architecture comprises a backbone network and three specialized branch networks, each serving distinct

**Table 1. KL and CKA values for the three fusion methods.**

|     | CCDFusion | PSFusion | Ours |
| --- | --- | --- | --- |
| KL | 1.073139 | 0.951814 | 0.873252 |
| CKA | 0.4292 | 0.4593 | 0.4793 |

functional purposes. The backbone network integrates LT with ResNet34 to hierarchically extract multi-level feature information from shallow to deep layers. This architecture is further enhanced by a superficial information fusion module (SIFM), which effectively integrates basic image information across different modules. The first branch network employs an invertible neural network (INN) for hierarchical detail information extraction, building upon the backbone network's output. This branch incorporates two specialized modules: an INN-based specialized detail fusion module (SDFM) and a multi-attention based deep detail fusion module (D2FM), which collectively facilitate the integration of both shallow and deep layer detail information. The second branch network, designated as the feature fusion network, performs layer-wise integration of basic and detail information through a deep feature fusion module (DF2M). This network implements bottom-up feature information aggregation utilizing the semantic injection module (SIM), while enhancing the layer output of LT to achieve optimal fusion results. Additionally, it strengthens the original scene fidelity path to enable accurate image reconstruction from feature information. The third branch network serves as a bottom-up semantic information extraction network, which progressively aggregates deep detail extraction information and processes it through the sparse semantic perception module (S2PM). This network combines convolutional layers with batch normalization (BN) and ReLU activation to achieve efficient feature-level semantic information extraction. Notably, our semantic extraction is implemented through the detail extraction sub-network for information aggregation, eliminating the need for a dedicated semantic feature extraction network. This innovative approach significantly reduces network parameters and computational overhead. Through extensive data training and parameter optimization in the PyTorch environment, we have thoroughly validated the effectiveness of our proposed method. As demonstrated in Fig 1e, our fusion network surpasses existing state-of-the-art (SOTA) networks in both detail preservation and visual perception, achieving superior performance metrics.

The two calibrated input images are jointly processed by our network, where each level of the backbone independently extracts multi-scale features from both images without shared parameters, which are dynamically optimized based on the global network structure. The backbone generates six hierarchical feature representations for each input: features from levels 1–4 are integrated by the SIFM to capture structural information, while levels 2–6 undergo detail refinement through the INN+SDFM and D2FM. The fused shallow and deep features are combined via hierarchical addition and further optimized by the DF2M, with the SIM and LT then collaboratively decoding the unified representation to produce the final fused image. This work presents several key advantages:

(1) Network architecture design: Our proposed framework employs a hierarchical architecture comprising a top-down backbone network integrated with bottom-up information processing pathways. The backbone network synergistically combines LT and ResNet architecture to enable comprehensive extraction of both basic and detailed features. Additionally, we have developed two specialized bottom-up networks for hierarchical information integration, incorporating advanced semantic information extraction mechanism to enhance feature representation and processing efficiency.

(2) Backbone network construction: We have optimized the original CNN and Transformer blocks to facilitate both local and global feature extraction while maintaining computational efficiency. The architecture incorporates coarse and LT modules as preprocessing components within ResNet, specifically integrating the LT network into the first two layers of ResNet. This design strategy enables comprehensive extraction of fundamental image information through the Transformer's multi-head attention mechanism, followed by progressive extraction of detailed texture features using the INN in a layer-wise manner.

(3) Auxiliary network architecture: Our framework incorporates a hierarchical feature integration system comprising three specialized modules. The superficial information fusion module (SIFM) facilitates the integration of fundamental features extracted from multiple network layers. For enhanced detail processing, we have developed two complementary components: The specialized detail fusion module (SDFM) and the deep detail fusion network (D2FM) synergistically facilitate an exhaustive integration of both superficial and profound details. The integrated base and detail features undergo layer-wise summation before being processed by our novel deep feature fusion module (DF2M), which

leverages multi-attention mechanisms for advanced feature re-integration. Furthermore, to substantiate the network's semantic extraction capabilities, we have implemented a bottom-up semantic information extraction pipeline that operates concurrently with the detail information consolidation process.

(4) Experimental validation: Through comprehensive experimentation and rigorous evaluation, we have validated the efficacy and robustness of our proposed network architecture. Quantitative and qualitative analyses demonstrate that our framework achieves superior performance in both fundamental feature aggregation and intricate detail preservation. Comparative studies against contemporary SOTA fusion networks reveal statistically significant improvements across multiple evaluation metrics, substantiating the competitive advantages of our approach.

The experimental results substantiate that our proposed network architecture achieves three critical objectives: (1) enhanced target saliency, (2) superior texture information preservation, and (3) comprehensive semantic feature extraction. The remainder of this paper is systematically organized as follows: Section 2 provides a comprehensive review of current IVIF methodologies and establishes the theoretical foundation for our proposed framework. Section 3 presents the architectural overview of our approach, accompanied by detailed technical explanations of each component. Section 4 conducts extensive experimental validation through comparative analysis with SOTA methods, employing both quantitative metrics and qualitative visual assessments. Additionally, we perform rigorous ablation studies to validate the architectural rationality and operational efficacy of our framework. Finally, Section 5 concludes the paper by summarizing our key contributions and discussing potential future research directions.

## 2. Preliminary

In recent years, deep learning-driven fusion networks have witnessed remarkable advancements, emerging as a predominant research focus in the field of image processing. This section presents a systematic review of the historical progression and technological evolution in deep learning-based image fusion methodologies. Furthermore, it establishes the theoretical foundation essential for our proposed framework, encompassing two critical components: (1) advanced encoding-decoding mechanisms based on Transformer architectures, and (2) fundamental theoretical principles of INN. These elements collectively form the cornerstone of our innovative approach to image fusion.

### 2.1. Current status of IVIF

The field of image fusion has undergone significant paradigm shifts, progressing from rudimentary pixel-level fusion to sophisticated transform-domain based methodologies. This technological evolution has witnessed the development of numerous advanced transform domain tools, including but not limited to DWT [12], NSST [13], and various adaptive filters with their derivatives [14–18]. Concurrently, researchers have pioneered several innovative fusion frameworks, such as sparse representation-based approaches [19–20], Markov random field models [21], and low-rank representation techniques [22,23]. These methodologies have dominated the image fusion landscape for decades, delivering remarkable innovations and enhanced visual performance. Nevertheless, the field currently faces substantial challenges in achieving groundbreaking advancements, presenting both obstacles and opportunities for contemporary academic research.

The advent of machine learning has catalyzed a transformative shift in image fusion methodologies, enabling researchers to transcend the limitations of conventional algorithms and explore innovative fusion strategies through deep learning paradigms. This transition has yielded remarkable outcomes and demonstrated unprecedented developmental potential [24]. Early deep fusion architectures primarily employed basic convolutional layers to extract image features [25–27], with fusion achieved through dimensionality reduction of these feature outputs. Notable contributions include Long *et al.*'s [28] residual dense network framework and Ma *et al.*'s [8] ResBlock-based dense-block networks, which incorporated shared encoding-decoding mechanisms and saliency detection to enhance feature learning. These pioneering architectures [25–30], characterized by their encoder-decoder structures with convolutional dense blocks, offered simplicity

and computational efficiency. However, researchers gradually recognized the inherent limitations of basic convolutional processing in capturing comprehensive image details, compounded by training process variability, often resulting in detail loss, texture degradation, and image blurring. This realization spurred the development of adversarial-based fusion networks [31–35], where the generator-discriminator paradigm was employed to enhance feature extraction. Despite their initial promise, these architectures revealed practical limitations, prompting further innovation. Liu *et al.* [33] pioneered a goal-driven dual adversarial learning network for multi-scene and multi-modal image fusion. Zhou *et al.* [34] enhanced adversarial networks by incorporating gradient and intensity components, developing a dual-discriminator architecture with distinct optimization objectives. Xu *et al.* [35] advanced the field through spectral and spatial loss-constrained adversarial networks. These evolutionary developments demonstrate that adversarial networks augmented with specialized feature representation elements significantly improve algorithmic adaptability and offer novel perspectives for fusion framework design.

The remarkable success of Transformer networks in natural language processing has catalyzed their adaptation for image fusion tasks, despite the computational challenges inherent in their architecture. Researchers have made significant strides in optimizing Transformer models for efficient image fusion. Ma *et al.* [36] pioneered a generalized fusion framework utilizing the Swin Transformer architecture, implementing an innovative long-short distance learning mechanism to enhance the multi-head attention process. Zhao *et al.* [10] advanced this approach through a dual-branch feature extraction network based on LT, incorporating specialized architectures for base and detail feature extraction. The integration of downstream vision tasks has emerged as a promising direction in image fusion research. Numerous studies have successfully embedded critical computer vision tasks (including image segmentation, object detection, and semantic segmentation) within fusion frameworks [11,37–40], creating synergistic relationships between fusion quality and task performance. Tang *et al.* [37] and Zhang *et al.* [38] implemented post-fusion semantic segmentation and object detection respectively, employing customized loss functions to establish mutual constraints. However, these approaches were limited to pixel-level semantic extraction. A significant advancement was achieved by Tang *et al.* [11], who reimagined high-level vision task integration by embedding semantic extraction within the fusion network itself, enabling feature-level semantic information extraction and substantially improving outcomes. Wang *et al.* [40] further innovated with an interactive enhancement paradigm that incrementally integrates saliency-based IVIF with object detection through layer-wise feature incorporation. The rapid evolution of deep learning-based fusion frameworks and their expanding application domains [41–45] underscore the tremendous potential for future research in this dynamic field. These developments not only demonstrate the adaptability of Transformer architectures but also highlight the growing sophistication of task-integrated fusion approaches.

However, existing deep learning-based fusion frameworks exhibit several notable limitations that warrant further investigation. The primary concern lies in the architectural design of early CNN-based fusion networks. These frameworks demonstrate excessive homogeneity, predominantly focusing on optimizing convolutional kernel designs and layer configurations, which consequently leads to substantial loss of detailed information in cross-modality feature representation. Although recent advancements integrating CNNs with Transformer architectures and downstream tasks have shown promising results, they simultaneously incur significantly increased computational costs and training complexity. Furthermore, the backbone networks designed for semantic information extraction often exhibit oversimplified structures with inadequate refinement. Another critical limitation is the insufficient theoretical foundation underlying CNN mechanisms. Current literature rarely provides comprehensive explanations regarding the internal operational principles of CNNs, nor does it adequately justify the rationale behind network architecture designs or the selection criteria for convolutional layer configurations. This theoretical gap hinders the development of more efficient and effective fusion frameworks. From a practical perspective, CNN-based networks face challenges in information preservation during forward propagation, particularly in balancing the extraction of basic and detailed features. This imbalance frequently results in fused images with weak detail contrast and sub-optimal visual quality. Additionally, the training process of most fusion networks

is constrained by limited dataset availability and insufficient sample diversity, coupled with a lack of robust theoretical guidance in training methodology development. In this study, we aim to systematically address these limitations through a comprehensive approach that encompasses architectural innovation, theoretical analysis, and practical implementation improvements in existing fusion algorithms.

## 2.2. Transformer and its variants

In recent years, the Transformer model [46] has emerged as a fundamental tool in natural language processing (NLP), showcasing exceptional feature extraction capabilities across both low-level [47–49] and high-level visual tasks [50–53], with significant practical applications. Building on its success, researchers have extensively explored and enhanced the Transformer model by leveraging its remote dependency mechanisms, leading to the development of innovative and efficient fusion networks such as SwinFusion [36], IFT [54], and AFT [55]. These advanced architectures have demonstrated outstanding performance in various domains, including classification tasks [56,57], target detection [58,59], image segmentation [60,61], and multi-modal learning [62,63]. Despite its remarkable achievements, the Transformer model is not without limitations. Its high computational demands and substantial hardware requirements pose significant challenges for practical deployment. To mitigate these issues, Wu *et al*. [64] introduced the Lite Transformer, which incorporates long-short range attention mechanisms and a planarized feed-forward network. This approach enables effective modeling of both global and local contextual information while significantly reducing the number of parameters, thereby maintaining the performance of the original Transformer with improved efficiency. In this work, we adopt the LT as the core algorithm for our framework, considering its multi-head attention mechanism and efficient parameter utilization. This choice not only facilitates the deep extraction of image feature information but also optimizes computational efficiency, making it well-suited for our objectives.

## 2.3. Invertible neural networks (INN) and its variants

Invertible neural networks (INN) has gained significant attention due to its distinctive feature extraction capabilities and ability to preserve information losslessly. These properties have enabled their successful application in various practical domains, including image coloring [65], information hiding [66], and high-resolution image processing [67]. A key strength of INN lies in its ability to enhance backbone network performance while minimizing memory consumption, achieved through innovative inter-crossing convolutional management strategies. Recognizing these advantages, we employ INN in this work as the primary mechanism for detail extraction and feature fusion. This choice is particularly motivated by its effectiveness in facilitating the seamless integration of detailed features across different network layers, thereby addressing critical challenges in multi-level feature representation.

## 2.4. The data flow between modules

Our proposed network architecture integrates ResNet and LT modules to implement a hybrid information processing paradigm that combines top-down feature extraction with bottom-up feature integration. The processing pipeline begins by standardizing input images to a 256×256 resolution before feeding them into our multi-stage backbone network. This backbone consists of six sequentially connected modules: (1) initial coarse feature extraction, (2) LT processing, (3) ResNet34 residual blocks followed by LT, (4) ResB and LT processing, and finally (5–6) two additional ResNet34 residual blocks. Through this architecture, we observe progressively decreasing feature scales across the six modules, leading us to classify the first four modules' outputs as basic features and the last two as detailed features. The fusion process involves several key operations: SIFM integrates basic features from all four initial modules, while INN performs coarse-to-fine detail extraction on modules 2–4 outputs. Subsequently, SDFM and D2FM handle detail integration for INN-processed features and final module outputs respectively. To address scale discrepancies, we implement an upsampling-based feature recombination strategy where integrated detail features are progressively combined with corresponding fundamental

features at each level. The complete fusion is ultimately achieved through DF2M module processing under SIM module coordination, which effectively combines the multi-scale feature representations to generate the final output image. This hierarchical approach ensures comprehensive utilization of both fundamental and detailed visual information throughout the fusion pipeline.

## 3. The proposed fusion network

In this chapter, we present our comprehensive conceptualization of the proposed network and offer a detailed explanation and illustration of its three main components. To enhance the persuasiveness of our work, we provide a comparative analysis that encompasses both theoretical aspects and the presentation of results. Additionally, we introduce the mechanism and principles behind the construction of the loss function, drawing on established theories in image fusion to seek insights and make improvements.

### 3.1. Analysis of the fusion framework

Deep learning has evolved a distinctive network architecture system through years of exploration and refinement. From the perspective of image fusion intrinsic architecture, nearly all deep learning networks are built upon CNNs, which rely on sufficient and reliable training sets as their foundational support. This enables computers to simulate the human brain, achieving high efficiency and intensive learning capabilities that surpass human abilities. Upon examining deep learning fusion frameworks over the past decade, it is evident that differences in fusion performance are primarily influenced by the design of the internal structure of the framework. The reasonableness and timeliness of this internal structure are crucial for effective information extraction. To highlight the differences in framework design, this paper focuses on two recent network architectures relevant to our study, which are briefly summarized in Fig 2 below.

In 2023, Zhao *et al*. [10] proposed the mutually driven two-branch feature decomposition network, CCDFuse, to emphasize both basic and detailed information. This network utilizes improved LT blocks and INN for targeted extraction of basic and detailed features, adopting an additive strategy to aggregate information features. Meanwhile, L. Tang *et al*. [11] revisited high-level vision tasks by implementing different layers of information extraction from the

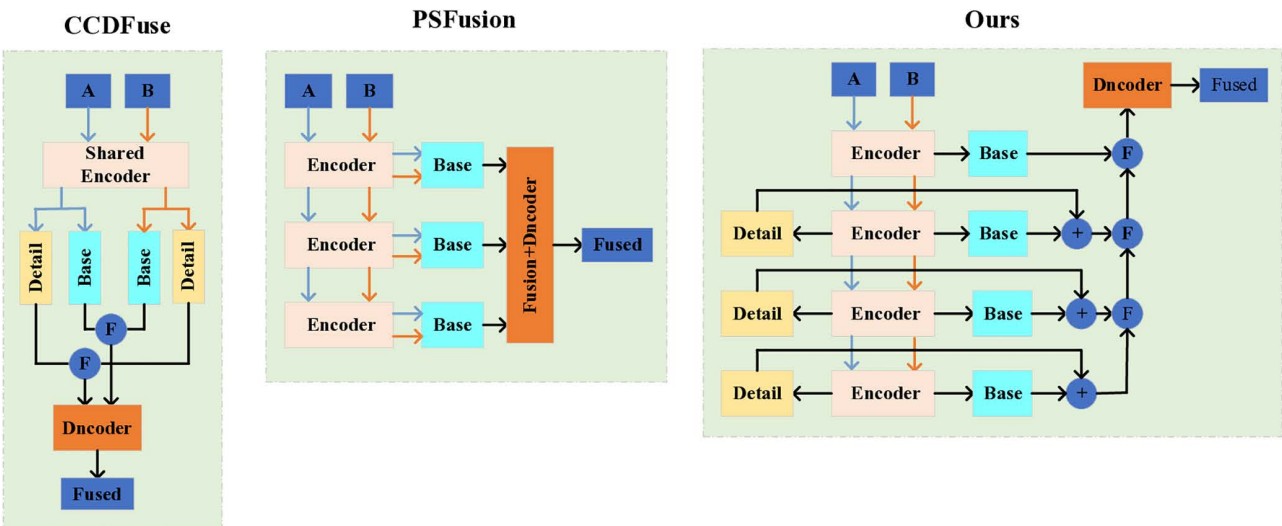

**Fig 2. A comparative analysis of streamlined fusion frameworks.**

input image based on ResNet. They then designed corresponding strategies to realize the network PSFusion for feature integration and semantic information extraction. A thorough review of the literature reveals that these two networks are considered advanced frameworks in the field of image fusion. Both frameworks demonstrate the advantages of current deep learning, whether through their overall design or sub-network architectural choices. However, they also exhibit some minor flaws. For instance, while CCDFuse has a commendable initial intention, its implementation is somewhat rudimentary. It forcibly uses the Restormer block and INN block in a loop to attempt basic and detailed information extraction, followed by a simple additive strategy to merge common and private features of the two modal images into the Restormer block for decoding, resulting in the fused image. On the other hand, PSFusion's overall framework design is both perfect and exquisite, fully considering the semantic requirements of downstream tasks. However, It neglects the private feature extraction of the two modal images throughout the network. Although the visual effect of fusion is highly satisfactory, the extraction of detailed information in regions rich in information features still requires improvement.

From the comparative framework diagram in Fig 2, it is evident that the CCDFuse framework is designed to be straightforward and user-friendly, yet it falls short in extracting deeper layers of information. In contrast, the PSFusion framework boasts a meticulous design that meticulously accounts for feature extraction of basic information across various levels, albeit without distinct separation of basic and private information. Drawing on the strengths and addressing the shortcomings of these two frameworks, we propose an optimized and enhanced scheme. Our objective is to facilitate the thorough extraction of both detailed and basic information across different layers, succeeded by the efficient amalgamation of this information through inter-layer integration.

## 3.2. Analysis of semantic segmentation framework

Semantic segmentation represents a sophisticated downstream task in image processing, extending beyond mere image classification. It constitutes a comprehensive prediction challenge within computer vision systems, with extensive applications in areas such as intelligent recognition, autonomous driving, and artificial intelligence. Initially, semantic segmentation methodologies predominantly relied on fully convolutional network architectures, utilizing mainstream classification backbones for image segmentation. Nonetheless, the swift advancement of deep learning networks has spurred the development of more efficient segmentation networks, including SegFormer [51], BANet [68], and SegNeXt [69]. Recently, Liu et al. [70] accomplished multi-modal image task segmentation employing the advanced SegFormer network, which stands as a cutting-edge algorithm in the semantic segmentation domain. It is noteworthy that numerous scholars have addressed the dual requirements of image fusion and semantic segmentation by integrating diverse feature elements into the model framework design, aiming for a synergistic effect between feature fusion and semantic segmentation modules. However, the majority of existing approaches position semantic information extraction subsequent to the fusion network [37,38], leading to uni-modal semantic information segmentation of the fusion outcome, which frequently falls short of precise semantic segmentation. Progressively, scholars have refined the original fusion framework design [11], transcending pixel-level feature extraction to embed semantic information extraction within the fusion framework. This entails the construction of a multi-branch backbone network to facilitate feature-level semantic information integration. Earlier methodologies depended entirely on the fusion result for semantic information extraction, with the fusion result's quality constrained by the fusion network's design, complicating the achievement of accurate information feature classification. PSFusion [11] re-examines semantic information features, eschewing the prior pixel-level semantic segmentation approach. Instead, it devises a specialized semantic segmentation network at the feature level, attaining high-accuracy segmentation. Drawing inspiration from PSFusion, our network amalgamates semantic information while sequentially integrating detailed features. This strategy seeks to fulfill the dual objectives of feature extraction and semantic segmentation without imposing additional parameter load.

## 3.3. Overall framework

Through a comprehensive analysis of the CCDFuse, PSFusion, and other network frameworks, we have identified inherent limitations in each. Our objective is to refine and enhance these frameworks to the greatest extent possible, striving for robust aggregation of information features and the optimal presentation of visual effects. Our network maintains the overarching structure of the PSFusion framework while incorporating LT sub-networks within the ResNet34 architecture to facilitate based information extraction across various levels. The INN network is utilized to intersperse the extraction of detailed features at multiple levels, ensuring comprehensive extraction of both basic and detailed information at each stage. Furthermore, we have developed several sub-networks to aid in the progressive integration of information at every level. The overall network architecture comprises a backbone network and three branch networks, enabling a top-down approach to information extraction and a bottom-up strategy for information integration. The specific network framework has been streamlined based on extensive experimentation, as depicted in Fig 3–5.

The overall framework is not a product of hasty design but rather the culmination of persistent exploration and refinement, achieved through rigorous data training and bolstered by SOTA computer hardware. Below, we provide a detailed description of the backbone and branch networks.

Figs 6 and 7 demonstrate the encoding-decoding pipeline for infrared and visible light image fusion in our proposed network. The framework processes the input images through parallel streams in the backbone network, effectively extracting both fundamental and detailed features, which are subsequently integrated via dedicated fusion modules. Notably, Fig 7 adopts UNet-like architecture that facilitates hierarchical feature aggregation through its bottom-up pathway. The final fused image is reconstructed through the LT decoder's output layer. The specific formulas for encoding and decoding will be elaborated in detail below.

**3.3.1. The backbone network.** As illustrated in Fig 3, the core structure of the framework is built upon LT and ResNet34. These two components work in tandem to extract the fundamental information from the input image,

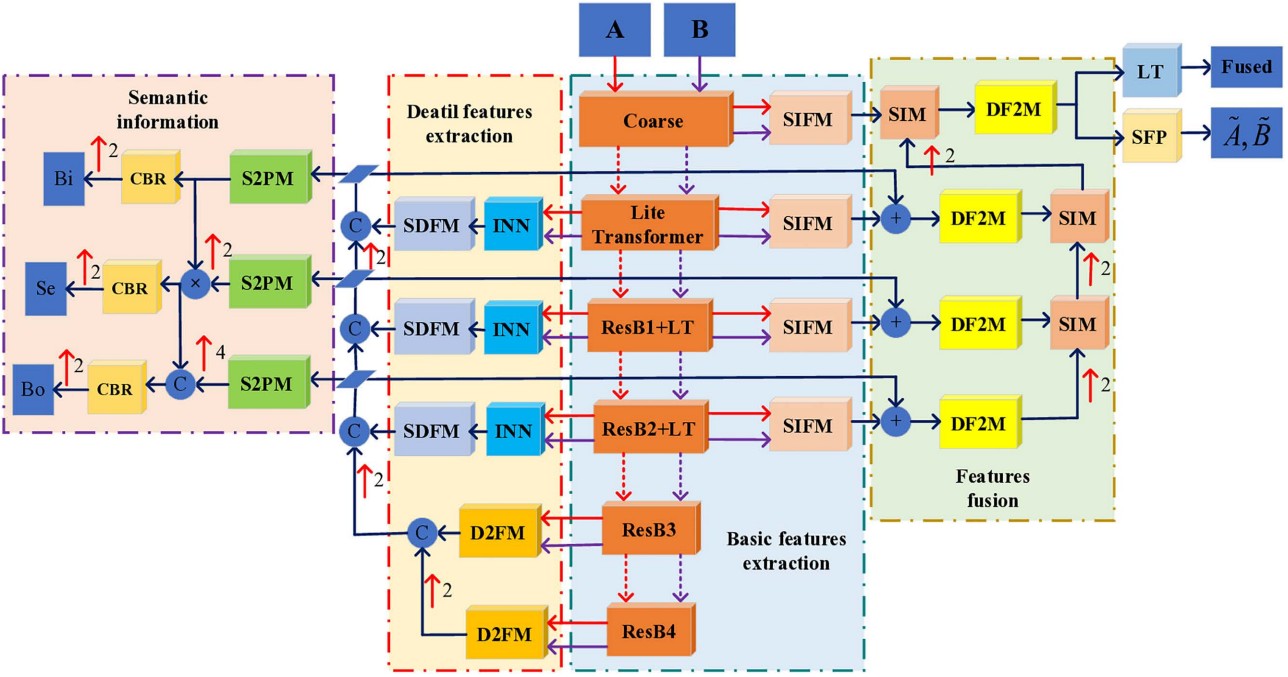

**Fig 3. Overall framework diagram.**

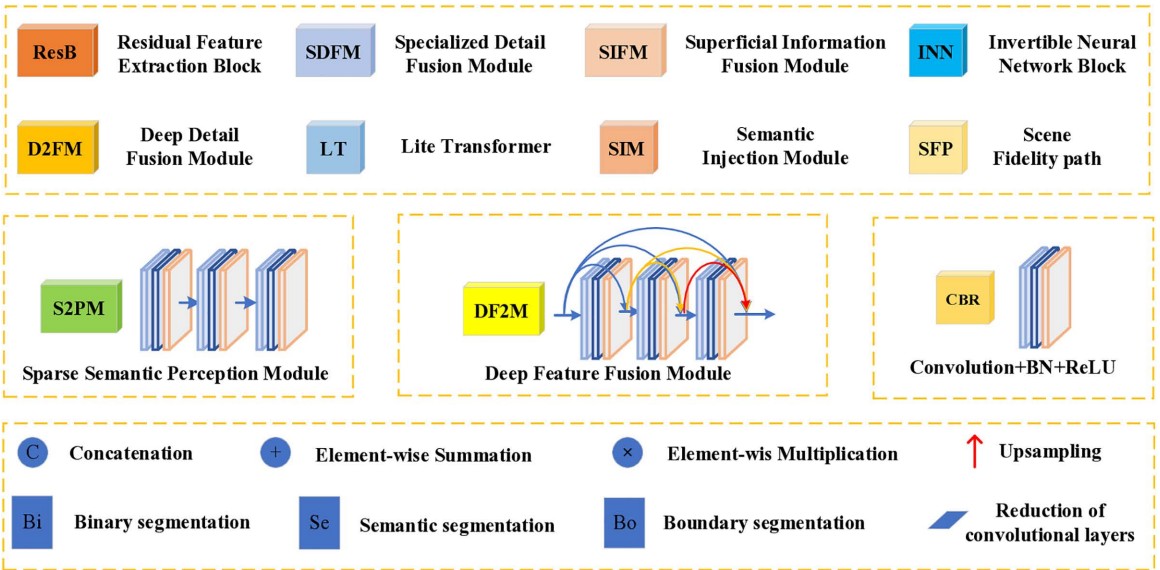

**Fig 4. Annotated supplementary documentation for the comprehensive framework schematic.**

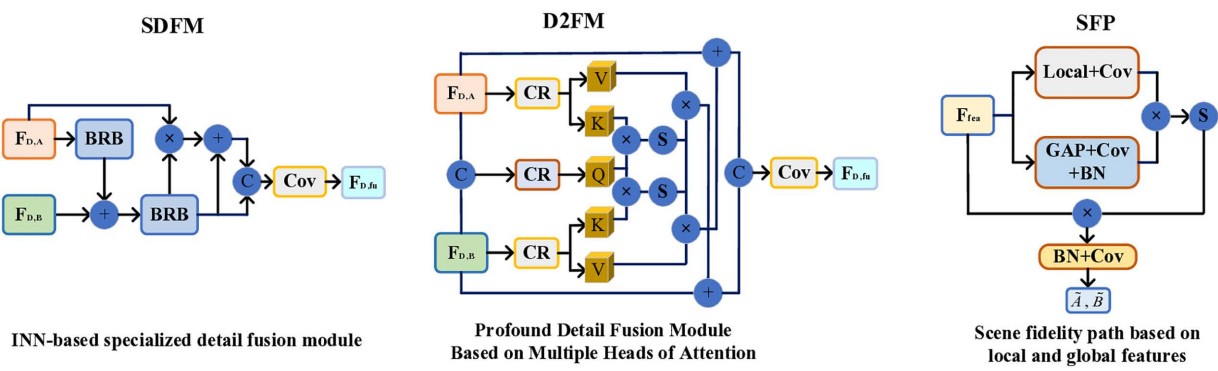

**Fig 5. A practical demonstration of key modules.**

facilitating a top-down, layer-by-layer information extraction process. Concurrently, the SIFM is engineered to effectively amalgamate the basic information extracted at each layer. Assuming a pair of aligned infrared $A \in R^{H \times W \times 1}$ and visible light $B \in R^{H \times W \times 3}$ images are provided, the initial step involves coarse-scale information extraction to derive the basic information of the first layer. In this structure, the coarse scale primarily consists of two $3 \times 3$ convolutions achieving preliminary information extraction from the original image, as encapsulated by the following formula:

$$Coarse_A\left(A\right) = FReLU\left(BN\left(Conv\left(FReLU\left(BN\left(Conv\left(A\right)\right)\right)\right)\right)\right),\tag{1}$$

$$Coarse_B\left(B\right) = FReLU\left(BN\left(Conv\left(FReLU\left(BN\left(Conv\left(B\right)\right)\right)\right)\right)\right),\tag{2}$$

$$F_A^1 = Coarse_A\left(A\right), F_B^1 = Coarse_B\left(B\right),\tag{3}$$

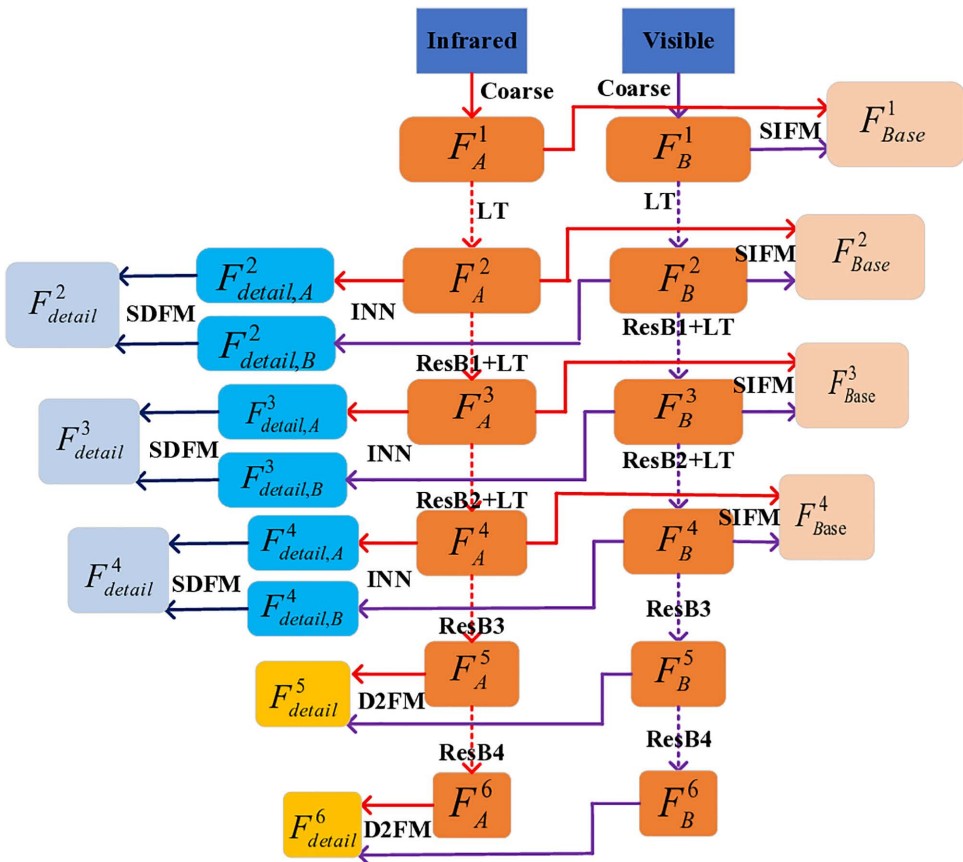

**Fig 6. IR & VIS image encoding flowchart.**

where BN represents the block normalization operation and *FReLU* stands for Flexible Rectified Linear Unit. The process initiates with a broad, shallow-level generalization and integration of the input information. Subsequently, the extracted coarse-scale information is channeled into the LT, where multiple attention heads collaboratively work to uncover the secondary layer of underlying information.

$$F_A^2 = LT\left(Conv\left(Conv\left(F_A^1\right)\right)\right), F_B^2 = LT\left(Conv\left(Conv\left(F_B^1\right)\right)\right).$$
(4)

For the internal construction of the LT block, please refer to reference [64]. The primary objective of this architectural design is to facilitate multi-angle extraction of basic features from the original image, leveraging the previously obtained coarse-scale information as a foundation. The processed information from the preceding layer is then sequentially propagated through the initial layer of ResNet32, which is strategically interleaved with the LT module, to progressively extract higher-level features corresponding to the third and fourth layers. This hierarchical feature extraction process can be formally expressed through the following mathematical representation:

$$F_A^3 = LT\left(\text{ResB1}\left(F_A^2\right)\right), F_B^3 = LT\left(\text{ResB1}\left(F_B^2\right)\right),$$
$$F_A^4 = LT\left(\text{ResB2}\left(F_A^3\right)\right), F_B^4 = LT\left(\text{ResB2}\left(F_B^3\right)\right),$$
(5)

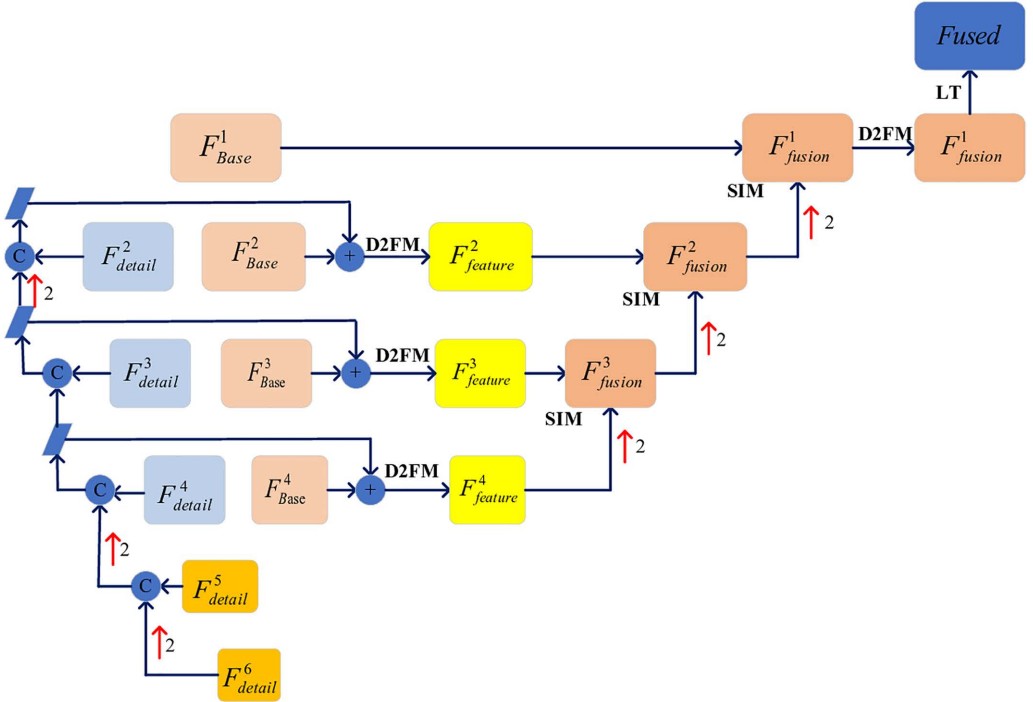

**Fig 7. IR & VIS image decoding flowchart.**

where *ResB*1, *ResB*2 denote the first two residual blocks of ResNet34. To capture more granular and sophisticated features, we employ direct feature extraction through the third and fourth modules of ResNet34, leveraging their inherent capacity for processing higher-level visual information. This strategic approach enables the network to progressively refine its feature representation, focusing on intricate patterns and detailed characteristics that are essential for comprehensive image understanding.

$$F_A^5 = \text{ResB}3\left(F_A^4\right), F_B^i = \text{ResB}3\left(F_B^4\right),$$
$$F_A^6 = \text{ResB}4\left(F_A^5\right), F_B^i = \text{ResB}4\left(F_B^5\right),$$

(6)

here $\left\{F_A^i, F_B^i\right\}, i = 1, 2, \cdots 6$ represent the hierarchical information features extracted from shallow to deep layers for infrared-visible images, respectively. *ResB*3, *ResB*4 represent the third and fourth residual blocks of ResNet34. A total of six levels of main information have been extracted based on the coordinated action of LT and ResNet34, of which the first four levels are the extraction of layer-by-layer general information of the original image, so we are extremely basic information here. The last two levels have penetrated into the interior of the information, which is memorized as the detail information.

As can be seen from Table 2, the original images are first unified into $256 \times 256$ and then fed into our network framework. The backbone network follows the principle of increasing the number of layers and decreasing the size.

**3.3.2. Base information integration.** The multi-level basic information captured by the backbone network is strategically fused with the corresponding SIFM to ensure comprehensive integration of essential details at each layer. This structured approach facilitates the seamless aggregation of overall information in subsequent stages. The fused base information can be expressed as:

$$F_{base}^i = SIFM\left(F_A^i, F_B^i\right), i = 1, 2, 3, 4.$$

(7)

**Table 2. Inputs and outputs at different levels.**

| | | Input | Output |
|---|---|---|---|
| 1 | Coarse | [4,1,256,256]/[4,3,256,256] | [4,32,256,256] |
| 2 | LT | [4,32,256,256] | [4,64,128,128] |
| 3 | ResB1+LT | [4,64,128,128] | [4,64,64,64] |
| 4 | ResB2+LT | [4,64,64,64] | [4,128,32,32] |
| 5 | ResB3 | [4,128,32,32] | [4,256,16,16] |
| 6 | ResB4 | [4,256,16,16] | [4,512,8,8] |

The SIFM employed in this study is the SDFM adopted from reference [11], which effectively enables both local and global information fusion. This module ensures optimal integration of multi-level features from infrared-visible images, maximizing the preservation and utilization of complementary information across different scales.

**3.3.3. Detail information integration.** Building upon the backbone network, we employ the INN architecture to facilitate hierarchical detail information extraction. To effectively integrate feature information across different network depths, we propose two novel modules: an INN-based SDFM and a D2FM enhanced with multi-attention mechanisms. The initial extraction of base-level detail information is accomplished through the INN framework, which can be formally expressed as:

$$F^i_{detail,A} = INN^{i-1}_A \left( F^i_A \right), F^i_{detail,B} = INN^{i-1}_B \left( F^i_B \right), i = 2, 3, 4, \tag{8}$$

$$F^i_{detail} = SDFM(F^i_{detail,A}, F^i_{detail,B}), i = 2, 3, 4, \tag{9}$$

INN primarily extracts the detailed depth information of infrared-visible images layer by layer. The specific framework structure can be referred to in literature [10]. SDFM is a detail integration module constructed based on the principles of information extraction from INN. The construction concept of SDFM is briefly explained as follows:

$$v^i_{detail} = BRB(BRB(F^i_{detail,A}) \oplus F^i_{detail,B}), i = 2, 3, 4, \tag{10}$$

$$F^i_{detail} = Cov(Cont(((v^i_{detail} \otimes F^i_{detail,A}) \oplus v^i_{detail}), v^i_{detail})), i = 2, 3, 4, \tag{11}$$

here, $F^i_{detail,A/B}, i = 2, 3, 4$ denote the detail information extracted from the two original images at different layers. $v^i_{detail}, i = 2, 3, 4$ represent the information combined in the SDFM process, while $F^i_{detail}, i = 2, 3, 4$ denote the detailed information integrated between the different layers. The BRB (bottleneck residual block) mainly refers to the structure described in [10], which balances the efficiency of computer operations and feature extraction capability. It consists of a sequential connection of 1×1 convolution+ReLU6, depth-wise convolution+ReLU6, and 1×1 convolution+Linear.

In conjunction with Fig 5, we designed D2FM to integrate deeper levels of detail information, with the aim of extracting deep detail information using multiple heads of attention. The specific formula is as follows:

$$\{K^i_A, K^i_B\} = Reshape \left\{ Cov(Cont(CR(F^i_A), CR(F^i_B))) \right\}, \tag{12}$$

$$\{V^i_A, V^i_B\} = Reshape \left\{ Cov(Cont(CR(F^i_A), CR(F^i_B))) \right\}, \tag{13}$$

$$Q = CR(Cont(F^i_A, F^i_B), \tag{14}$$

$$D_A^i = Softmax(K_A^i \otimes Q^i) \otimes V_A^i \oplus F_B^i, D_B^i = Softmax(K_B^i \otimes Q^i) \otimes V_B^i \oplus F_A^i, \tag{15}$$

$$F_{detail}^i = Cov(Cont(D_A^i, D_B^i)), i = 5, 6, \tag{16}$$

where CR denotes the joint convolution+ReLU operation on the feature information. Re*shape*( · ) denotes the reshape operation. *Cont*( , ) indicates that the front and back items are spliced. $K_{A/B}^i$, $V_{A/B}^i$, $Q_{A/B}^i$ denote the key, value and query of the two modal image information.

### 3.3.4. Feature integration network.

The inter-layer basic and detail information has been efficiently extracted and integrated in the preceding network through a layer-by-layer process. As depicted in Fig 4, we perform a hierarchical summation of the integrated basic and detail information, which is then fed into the DF2M. This process facilitates bottom-up feature information aggregation, guided by the semantic injection module (SIM). Ultimately, the layer output of LT is enhanced to achieve the final fusion result, while the original scene fidelity path (SFP) is refined to predict the original image from the feature information. The DF2M consists of three sequentially connected density blocks, as illustrated in Fig 4. To optimize the bottom-up information integration, we retain the framework structure proposed in literature [11].

$$D_{detail}^4 = Cov(Cont(F_{detail}^4, \uparrow Cont(F_{detail}^5, \uparrow F_{detail}^6))), \tag{17}$$

$$D_{detail}^3 = Cov(Cont(F_{detail}^3, D_{detail}^4)), \tag{18}$$

$$D_{detail}^2 = Cov(Cont(F_{detail}^2, \uparrow D_{detail}^3)), \tag{19}$$

$$F_{feature}^i = DF2M(F_{base}^i \oplus D_{detail}^i), i = 2, 3, 4, \tag{20}$$

$$F_{fusion}^3 = SIM(F_{feature}^3, \uparrow F_{feature}^4), F_{fusion}^2 = SIM(F_{feature}^2, \uparrow F_{feature}^3), \tag{21}$$

$$F_{fusion}^1 = SIM(F_{base}^1, \uparrow F_{fusion}^2), Fused = LT(DF2M(F_{fusion}^1)), \tag{22}$$

Here, $D_{detail}^i, i = 2, 3, 4$ denote the merging of the detailed information obtained from the integration of each layer into the previous three layers. $F_{feature}^i, i = 2, 3, 4$ represent the result of summing the base and detail information of each layer before sending it to DF2M for processing. To achieve lossless information transfer between layers, $F_{fusion}^i, i = 1, 2, 3$ denote the results of integrating inter-layer information using SIM, and *Fused* denotes the final fusion output.

To harmonize the architecture of the entire network, we improved the SFP model from literature [11] to derive the prediction results of the source images. As illustrated in Fig 5, the refined framework structure is as follows:

$$Local = Cov(Local(F_{fea})), Global = BN(Cov(GAP(F_{fea}))), \tag{23}$$

$$\tilde{A}/\tilde{B} = Cov(BN(Softmax(Local \otimes Global) \otimes F_{fea})), \tag{24}$$

where $Local(\cdot), GAP(\cdot)$ denote local and global processing of feature information respectively. BN denotes the block normalization operation and $\tilde{A}/\tilde{B}$ denote the final prediction results for the input images.

### 3.3.5. Semantic information integration network.

The semantic information extraction network is designed without the need for a separate framework system; instead, it is implemented through the progressive upward aggregation of deep detail information across layers. Initially, the integrated detail information is processed by the sparse semantic perception module (S2PM), which performs deep convolutional operations on the information features. Subsequently, a straightforward top-down information aggregation network is constructed. Finally, the extraction of binary, semantic, and boundary information is accomplished using a combination of convolution, batch normalization, and ReLU activation (CBR). As shown in Fig 4, the S2PM comprises a dense block network formed by three sequentially connected CBR layers.

$$Binary = \uparrow CBR(S2PM(D_{detail}^1)), \tag{25}$$

$$Semantic = \uparrow CBR(S2PM(D_{detail}^1) \otimes \uparrow S2PM(D_{detail}^2)), \tag{26}$$

$$Boundary = \uparrow CBR(Cont((S2PM(D_{detail}^1) \otimes \uparrow S2PM(D_{detail}^2)), S2PM(D_{detail}^3)), \tag{27}$$

where *Binary*, *Semantic* and *Boundary* denote binary, semantic and boundary information, respectively.

## 3.4. Loss function

The proposed framework incorporates a fusion loss function that integrates intensity, gradient, and structural similarity metrics to directly constrain the fusion output results. Furthermore, auxiliary loss and reconstruction loss are employed to indirectly regulate the network's feature extraction and integration processes. In the following sections, we provide a detailed explanation of the fusion loss, auxiliary loss, and reconstruction loss.

### 3.4.1. Fusion loss.

We design corresponding loss functions based on intensity, gradient, and similarity to jointly constrain the fusion results, enabling continuous optimization of the output through the coordinated interplay between the fusion framework and the loss functions. As illustrated in Fig 1, the CCDFuse framework, despite its concise design, falls short of achieving high-quality fusion results due to the lack of strong constraints, such as gradient preservation in the fused images. To overcome this limitation, we propose a more robust fusion loss strategy built upon the CCDFuse framework and introduce corresponding parameters to fine-tune the final fusion output. The intensity loss function is defined as follows:

$$L_{int} = \frac{1}{HW} \| S(M_A + M_{tar}) \otimes (fu - A) \|_1 + \frac{1}{HW} \| S(M_B + (1 - M_{tar})) \otimes (fu - B) \|_1, \tag{28}$$

$$M_x = \begin{cases} 1, & if\, \sigma^2(x) > \sigma^2(y), \\ 0, & otherwise, \end{cases} \tag{29}$$

here, $M_x(x = A/B)$ denotes the contrast mask of the original image. $M_{tar}$ denotes the salient target mask, which aims to guide the fusion network to retain more targets in the infrared image and can be simply generated from the semantic segmentation labels. $y$ denotes another modality, $\sigma^2(\cdot)$ denotes the variance of the different modal images, and $fu$, denotes the fused image. The gradient loss function is defined as follows:

$$L_{grad} = \frac{1}{HW} \| |\nabla fu| - max(|\nabla A|, |\nabla B|) \|_1, \tag{30}$$

where $\nabla$ denotes the Sobel gradient computation, $|\cdot|$ is the absolute computation, and $max(\cdot)$ denotes the selection out of the element-wise maximum. The similarity loss function is defined as follows:

$$L_{ssim} = \frac{1}{SSIM(A, fu) + SSIM(B, fu)}, \tag{31}$$

$SSIM(,)$ denotes the structural similarity of the two images.

$$L_{fusion} = \alpha_1 L_{int} + \alpha_2 L_{grad} + \alpha_3 L_{ssim}, \tag{32}$$

where $\alpha_i, i = 1, 2, 3$ are used to balance the weights between different loss functions and regulate the final fusion loss.

**3.4.2. Auxiliary loss.** In the proposed fusion framework, multi-layer extraction and integration of base and detail information are conducted. To effectively constrain the fusion features and ensure that the fused image incorporates more informative features, we introduce a correlation loss based on inter-layer information.

$$L_{cc} = \frac{\sum_{i=1}^{4} \left(CC(F_{detail,A}^i, F_{detail,B}^i)\right)^2}{\sum_{j=1}^{3} \left(CC(F_A^i, F_B^i)\right) + \varepsilon}, \tag{33}$$

here, $CC(,)$ denotes the correlation coefficient operator, and $\varepsilon$ (set 0.01) is a constant ensuring that the denominator is a positive number. Additionally, the scene fidelity loss is increased to guarantee the efficiency of reconstructing the original image.

$$L_{SFP}^x = \frac{1}{HW}\|x - \tilde{x}\|_1 + \frac{1}{HW}\||\nabla x| - |\nabla \tilde{x}|\|_1, \tag{34}$$

in addition, to ensure sufficient semantic information output, reference [11] designs corresponding binary loss $L_{bi}$, semantic loss $L_{se}$ and boundary loss $L_{bo}$. The binary loss is designed using a weighted cross-entropy loss to mitigate the class imbalance between the object and the background. The semantic loss is implemented through OHEMCELoss, while the boundary loss is formulated by applying a cross-entropy loss function to measure the discrepancy between the predicted boundary results and the ground truth.

$$L_{ours} = \lambda_f L_{fusion} + \lambda_c L_{cc} + \lambda_{sf}(L_{SFP}^A + L_{SFP}^B) + \lambda_{se}(L_{se} + L_{bo} + L_{bi}), \tag{35}$$

where $\lambda_f, \lambda_c, \lambda_{sf}, \lambda_{se}$ are hyper-parameters used to regulate the fusion loss, relevance loss, scene fidelity loss and semantic relevance loss. The optimal values are found during training and simulation on a large amount of data.

## 4. Experimental argumentation

The previous section primarily outlined the overall framework structure of the proposed method and the construction of the loss function from a theoretical perspective. This section aims to supplement and further validate the theoretical foundations through empirical analysis. To comprehensively demonstrate the effectiveness of the proposed framework, we select several widely used infrared-visible datasets for training and testing, ensuring robust validation across multiple datasets. First, we detail the experimental configurations and determine the optimal parameters through extensive comparative analyses of experimental operations. Next, we conduct qualitative and quantitative evaluations on three distinct datasets to highlight the advantages of the proposed network, both intuitively and theoretically. Furthermore, we present semantic

segmentation results under different segmentation models to showcase the superior performance and generalization capability of our approach. Finally, to substantiate the design rationale of the proposed framework, we perform two sets of ablation experiments to underscore the rationality and necessity of its structural components.

### 4.1. Preparation for the experiment

The MSRS [37] dataset is utilized for training and validation of our model, while the RoadScene [27], MSRS, and M³FD [33] datasets are employed for testing. For the visualization and comparison experiments, we select three representative image pairs: a daytime image pair from RoadScene, a nighttime image pair from MSRS, and a haze image pair from M³FD, which features challenging external environmental conditions. To ensure a comprehensive and realistic evaluation of the fusion performance, we compare our method with nine classical deep learning-based fusion approaches. These include the generalized fusion framework IFCNN [26], based on convolutional neural networks; and the adversarial-based fusion network DDCGAN [32]. Additionally, we incorporate the widely recognized STDFusion [8], which leverages saliency detection; SeAFusion [37], a semantic-aware fusion network; and SwinFusion [36], a fusion network based on the Swin Transformer. We include several advanced networks proposed in 2023, such as the interactive reinforcement fusion network IRFS [40], which integrates saliency detection; the correlation-driven dual-branch fusion network CDDFuse [10]; and the progressive semantic injection-based fusion network PSFusion [11]. Additionally, to ensure methodological currency in our comparisons, we integrated the diffusion Transformer-based feature-guided image fusion framework recently developed by Yang *et al.* [71] (2025) (LFDT). This architecture demonstrates particular efficacy for multi-modal image fusion tasks while maintaining robust generalization capabilities. To evaluate the extraction of semantic information, we employ three classical segmentation models (BANet [68], SegFormer [51], and SegNeXt [69]) to qualitatively and quantitatively assess the fusion results on the MSRS dataset.

To comprehensively demonstrate the effectiveness of the proposed fusion networks, we employ six widely recognized statistical evaluation metrics. These include generalized quantitative assessment metrics such as entropy (EN) [72], standard deviation (SD) [73], mutual information (MI) [74], visual information fidelity (VIF) [75], sum of correlation differences (SCD) [76], and edge preservation ($Q_{AB/F}$) [77]. For all these metrics, higher values indicate superior fusion performance. Additionally, to evaluate the semantic segmentation results, we conduct qualitative and quantitative analyses on the MSRS and MFNet [78] datasets using the pixel intersection over union (IoU) metric, which is a standard evaluation tool in segmentation models.

The proposed framework is designed to generate high-quality fused visual images through the synergistic interaction of feature extraction and loss functions, while also emphasizing the extraction of semantic perceptual information. After extensive experimental training and empirical tuning, the hyper-parameters for the various loss functions are set to $\alpha_1 = \alpha_2 = \alpha_3 = 1, \lambda_f = 10, \lambda_c = 5, \lambda_{sf} = 5, \lambda_{se} = 10$ and $\varepsilon = 0.01$. The model is trained using the classical stochastic gradient descent (SGD) method with a batch size of 16. The learning rate is initialized at 0.001 and follows a decay strategy. The training process spans 2500 epochs over approximately 30 hours, ensuring that the intrinsic features of the images and the semantic information are thoroughly explored. All input images are normalized to a range between 0 and 1 to facilitate consistent processing within the network. Based on prior experience, the YCbCr color space is adopted for processing color images. The proposed network is implemented on the PyTorch platform using the PyCharm tool, while the other eight comparison networks and segmentation models are implemented as described in their respective original papers. All experiments in this study were conducted on an NVIDIA GeForce RTX 4080 GPU and a 13th Gen Intel(R) Core(TM) i7-13700F 2.10GHz processor.

### 4.2. Fusion comparison and analysis

Once the network training achieves robustness and stability, we conduct experimental comparisons and analyses on the RoadScene, MSRS, and M³FD datasets. To ensure a comprehensive and realistic evaluation, we select representative

scenarios from each dataset: a daytime scenario, a nighttime scenario, and a haze scenario characterized by challenging external conditions. In this section, we present both qualitative and quantitative results, enabling a dual validation of the proposed method through intuitive visual assessments and theoretical data analysis.

**4.2.1. RoadScene experiments.** Here, we test 50 pairs of infrared-visible images from the RoadScene dataset and select one pair of images from daytime scenes for visualization. The exact results of the experiment are shown in Fig 8, with a partially enlarged portion displayed in Fig 9. Given the substantial volume of image pairs in the RoadScene dataset, it is impractical to display each result individually. Consequently, we have adopted a cumulative distribution approach to effectively present the six key metric values derived from our experimental results, as illustrated in Fig 10. This methodological approach enables a comprehensive and statistically robust representation of our findings across the entire dataset.

By analyzing Figs 8 and 9 in combination, several key observations can be made regarding the performance of different fusion methods. IFCNN and DDCGAN exhibit inadequate integration of letter information in the red zoomed-in areas located at the lower left corner. STDFusion and SeAFusion present considerable advancements in ground marking integration; however, as revealed in Fig 9, these methods struggle with proper integration around vehicles and pedestrians, suffering from overexposure issues that result in less distinct target features. The IRFS image appears notably darker, while SwinFusion fails to achieve sufficient detail extraction. CDDFuse manifests two critical limitations: overexposure on the vehicle's side and seriously inadequate detail extraction. LFDT has a certain advantage in terms of clarity, but it still lacks in processing detailed information, particularly evident in the display of ground letters. Although PSFusion demonstrates substantial improvement over previous fusion methods, a detailed comparison with Fig 9 reveals that its detail extraction capability, particularly on the right side of the vehicle and around pedestrians, remains inferior to our proposed method. In contrast, our method successfully integrates the complementary information from source images while providing superior detailed information for characterization, outperforming all comparative methods in terms of both integration quality and detail preservation.

The RoadScene dataset comprises 50 carefully curated infrared-visible image pairs, encompassing diverse day and night scenarios, which were rigorously evaluated using cumulative distribution functions for comprehensive analysis. As illustrated in Fig 10, the performance analysis reveals distinct patterns among the evaluated methods: IFCNN, DDCGAN,

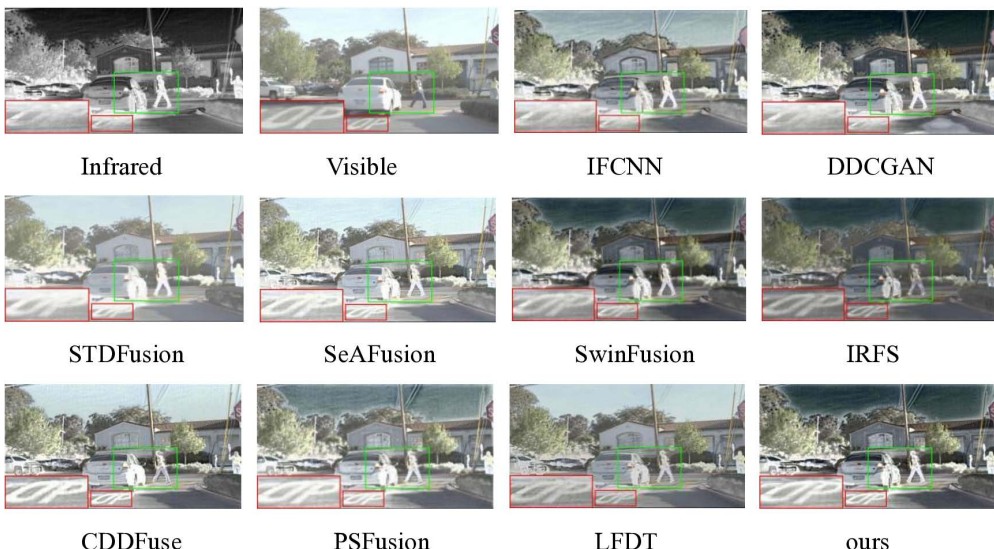

**Fig 8. Qualitative comparison of ten fusion methods on scene 05005 from the RoadScene dataset.**

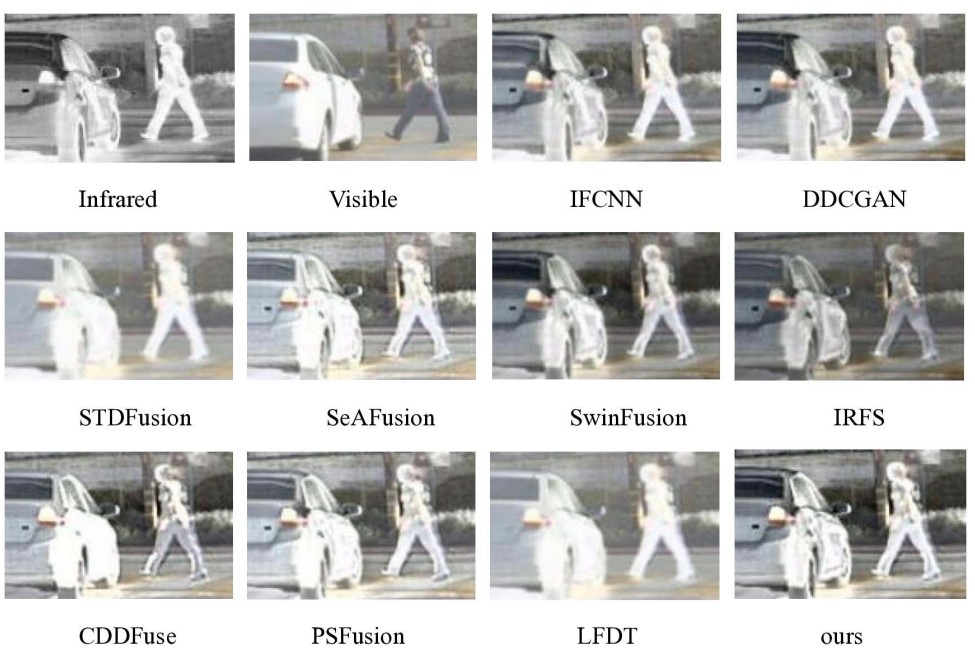

**Fig 9. Localized zoom effects for ten fusion results of the 05005 scene.**

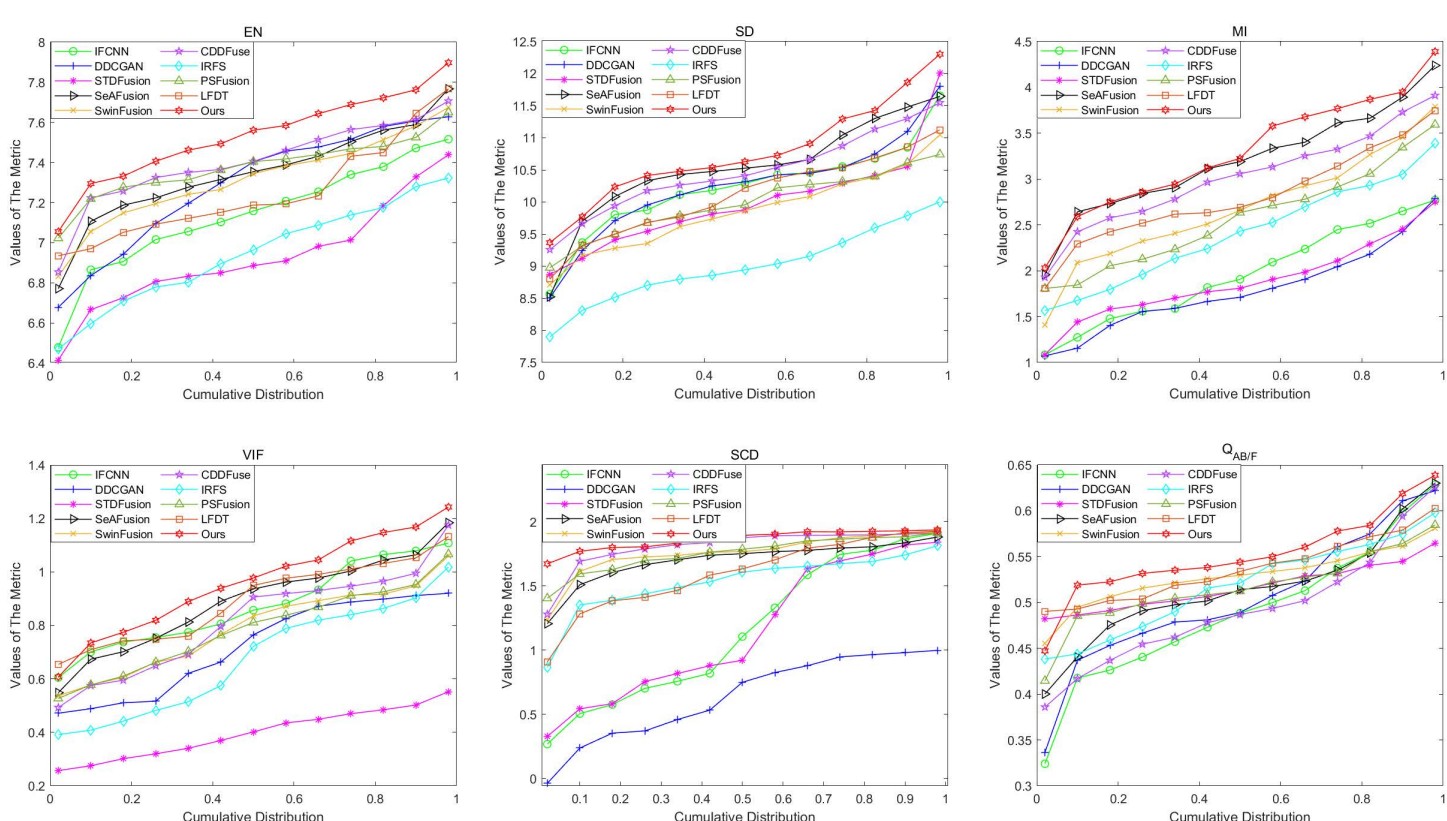

**Fig 10. Cumulative distribution of the six metrics on the RoadScene dataset.**

and STDFusion demonstrate suboptimal performance across all metric values. While SwinFusion and CDDFuse show relatively improved metric values, our proposed method consistently achieves superior performance, attaining the highest scores across all six evaluation metrics. Notably, our approach outperforms all competing methods except PSFusion and LFDT, which shows comparable but slightly inferior results. These extensive tests on the RoadScene dataset conclusively demonstrate that our method delivers highly satisfactory and robust performance across various environmental conditions.

**4.2.2. MSRS experiments.** For comprehensive evaluation, we conducted extensive testing on 361 infrared-visible image pairs from the MSRS dataset, with the experimental results systematically presented in Figs 11 and 12. Specifically, Fig 11 showcases a representative pair of late-night images, carefully selected to demonstrate the fusion performance under challenging low-light conditions, thereby highlighting the method's effectiveness across diverse environmental scenarios.

As illustrated in Fig 11, the late-night environment presents a particularly challenging scenario for image fusion, as both infrared and visible light sensors capture significantly less information compared to daytime conditions, thereby testing the robustness of fusion methods. The fusion results reveal distinct performance variations among the evaluated approaches: DDCGAN and STDFusion produce fuzzy and dark outputs. IFCNN, SwinFusion, IRFS, CDDFuse and LFDT demonstrate clear limitations in preserving target details and distant building features, with magnified targets appearing predominantly dim. Although PSFusion outperforms other methods in maintaining brightness and detail for both distant houses and nearby targets, it still falls short of our proposed method in target detail extraction. Consequently, our method demonstrates remarkable advantages in challenging low-light conditions, particularly in dark night environments, establishing its superior capability in handling adverse external conditions.

The cumulative distribution function analysis of 361 image pairs, as presented in Fig 12, clearly demonstrates the superior performance of our proposed method across multiple evaluation metrics. This comprehensive evaluation reveals that our method maintains consistent advantages in several critical aspects: detail information extraction, visual fidelity preservation, correlation maintenance between fused images, and mutual information retention.

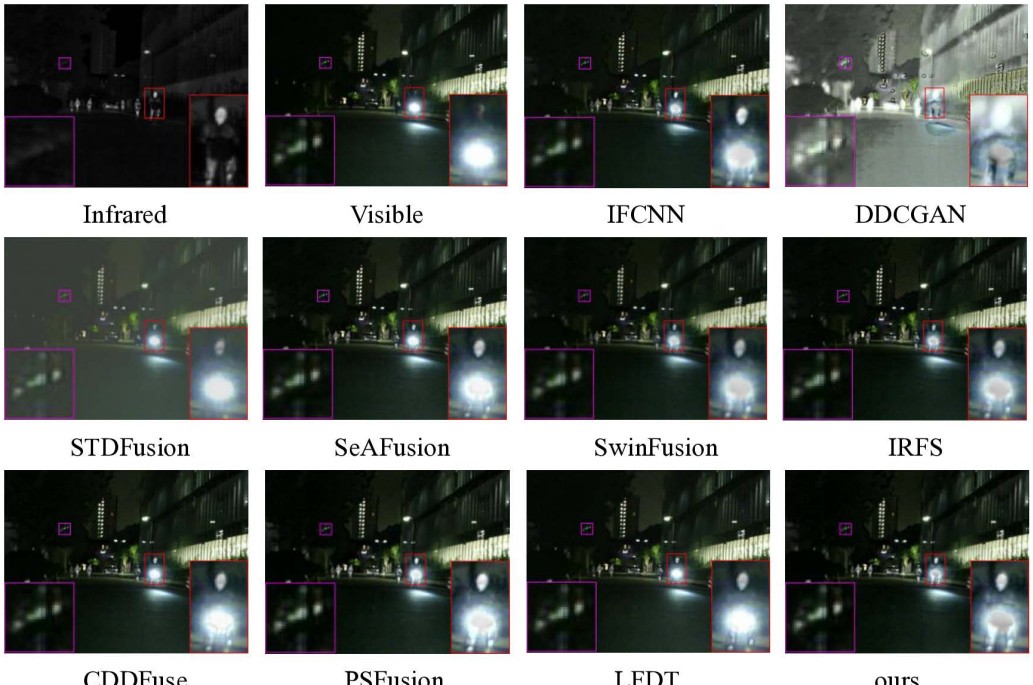

**Fig 11. Qualitative comparison of ten fusion methods on scene 00754N from the MSRS dataset.**

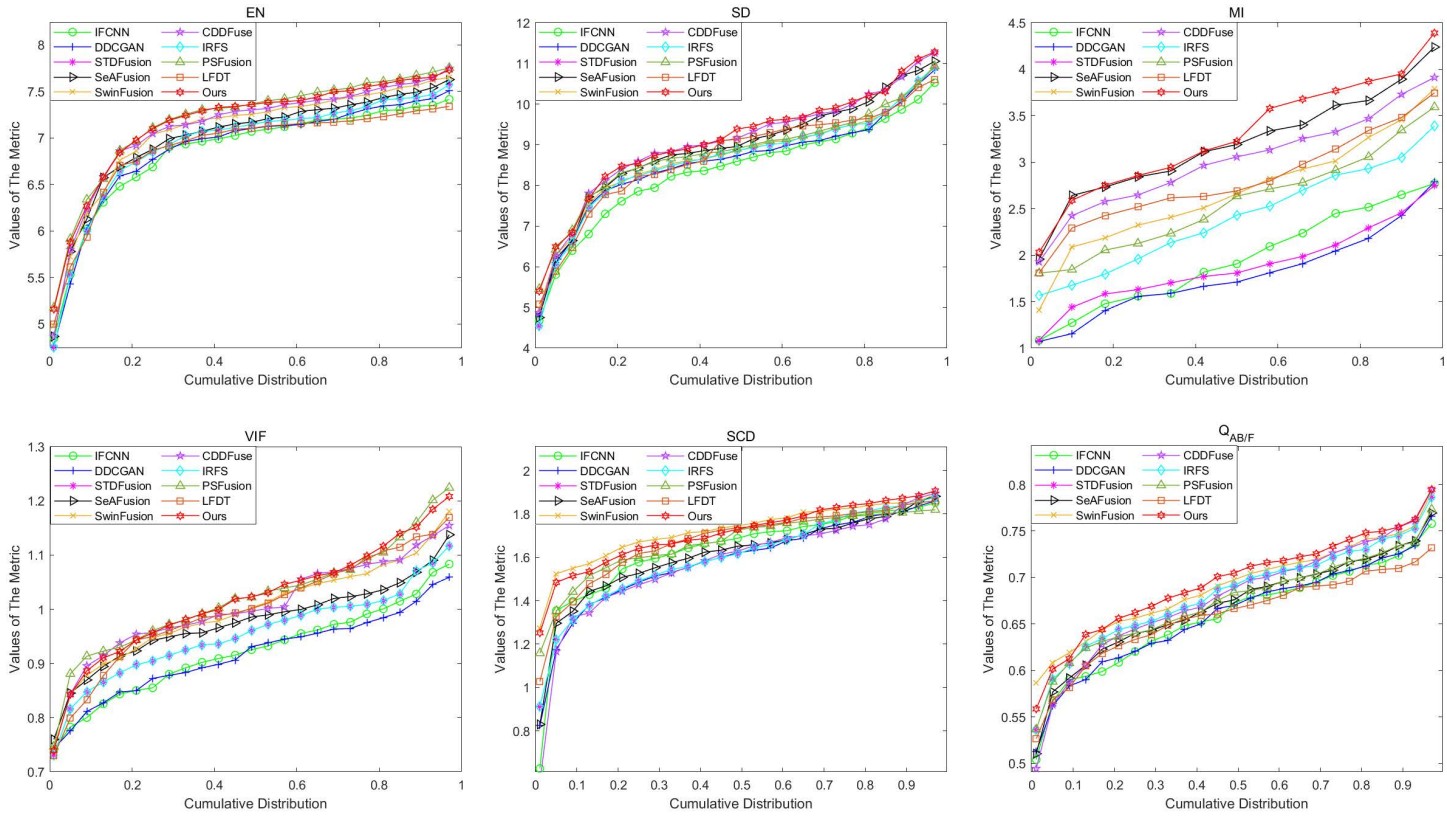

**Fig 12. Cumulative distribution of the six metrics on the MSRS dataset.**

**4.2.3. M³FD experiments.** The M³FD dataset consists of 300 pairs of outdoor infrared-visible images, predominantly captured under challenging environmental conditions including haze, rain, and low visibility scenarios. These adverse conditions significantly influence the image quality and characteristics during acquisition. For visual demonstration purposes, we have selected a representative set of images captured under severe haze conditions, which effectively illustrates the performance of fusion methods in extreme environmental situations.

As demonstrated in Fig 13, the comparative analysis reveals distinct performance characteristics among the evaluated methods. DDCGAN exhibits an extremely blurry condition. IFCNN exhibits a significant black spot artifact that compromises image quality. STDFusion, SeAFusion, and CDDFuse show limited effectiveness in haze removal, failing to achieve satisfactory de-fogging results. IRFS presents severe black shadow artifacts throughout the image. LFDT is still significantly affected by fog, and the brightness of the target objects is insufficient. Although SwinFusion and PSFusion show relatively better performance, they still demonstrate clear limitations in detail extraction capabilities. In contrast, our proposed method not only demonstrates effective haze reduction but also significantly outperforms other methods in preserving and enhancing detailed information, establishing its superior performance in challenging foggy conditions.

As illustrated in Fig 14, the proposed fusion method demonstrates superior performance across the majority of evaluation metrics. Notably, SwinFusion and PSFusion show comparable results to our method specifically in terms of VIF and SCD metrics. This performance similarity suggests that these two methods also achieve commendable results in maintaining visual fidelity and preserving correlation between source and fused images. However, our proposed method maintains an overall advantage across the comprehensive set of evaluation metrics.

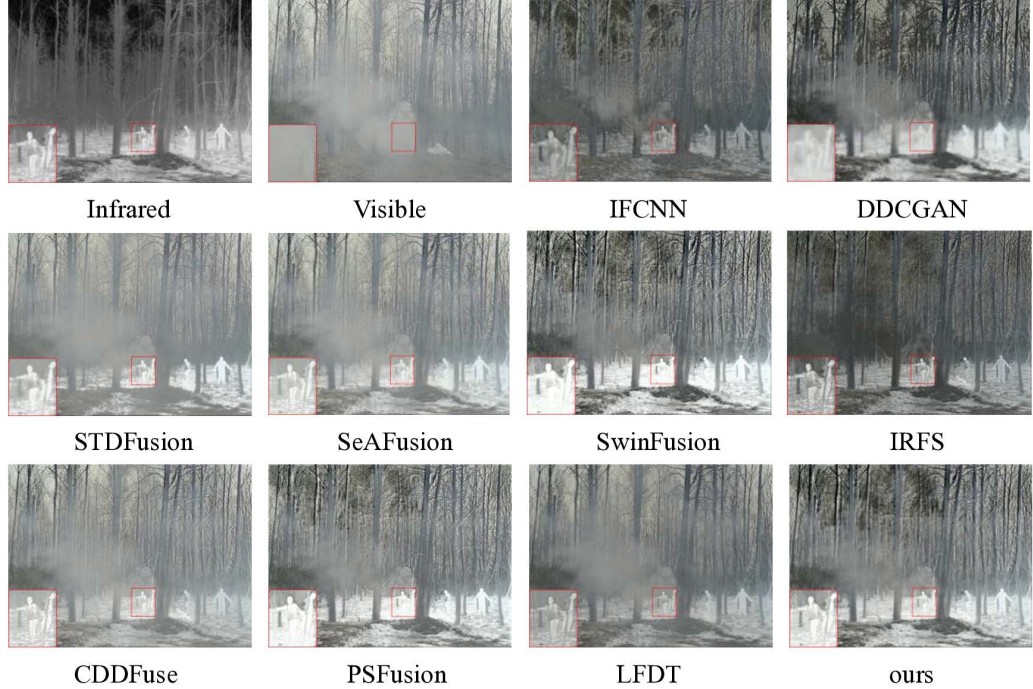

**Fig 13. Qualitative comparison of ten fusion methods on scene 01443 from the M³FD dataset.**

In conclusion, the proposed framework demonstrates superior performance in both luminance representation and detail extraction compared to existing methods. Notably, it exhibits robust performance even under challenging environmental conditions, a capability primarily attributed to the innovative design of its LT and detail sub-networks.

### 4.3. Segmentation comparison and analysis

Based on the comprehensive analysis of fusion experiments, we conduct a systematic evaluation of the semantic segmentation performance across ten fusion methods using the MSRS dataset. For the segmentation experiments, we employ the classical BANet architecture as the primary model, complemented by two state-of-the-art models, SegFormer and SegNext, to ensure robust validation. The quantitative and qualitative segmentation results are presented in Figs 15 and 16, respectively.

The semantic segmentation results of the ten fusion methods, as obtained through the classical BANet architecture, are presented in Fig 15. A comparative analysis reveals that our proposed method demonstrates superior performance in semantic information extraction, particularly evident in its enhanced capability to identify distant and challenging targets, including pedestrians and bicycles. This remarkable performance can be fundamentally attributed to the well-designed network architecture, which establishes a robust framework for effective semantic feature extraction. Specifically, the detail sub-network's sophisticated information integration mechanism plays a pivotal role in preserving and enhancing critical semantic features throughout the processing pipeline.

The quantitative segmentation results of the three models BANet, SegFormer, and SegNeXt on the MFNet dataset are illustrated in Table 3. As evidenced by the statistical data presented in Table 3, our proposed method consistently achieves superior performance, obtaining the highest IoU values across critical categories including car, bike, and person, while maintaining competitive scores in other categories. Notably, all three segmentation models attain their peak average IoU

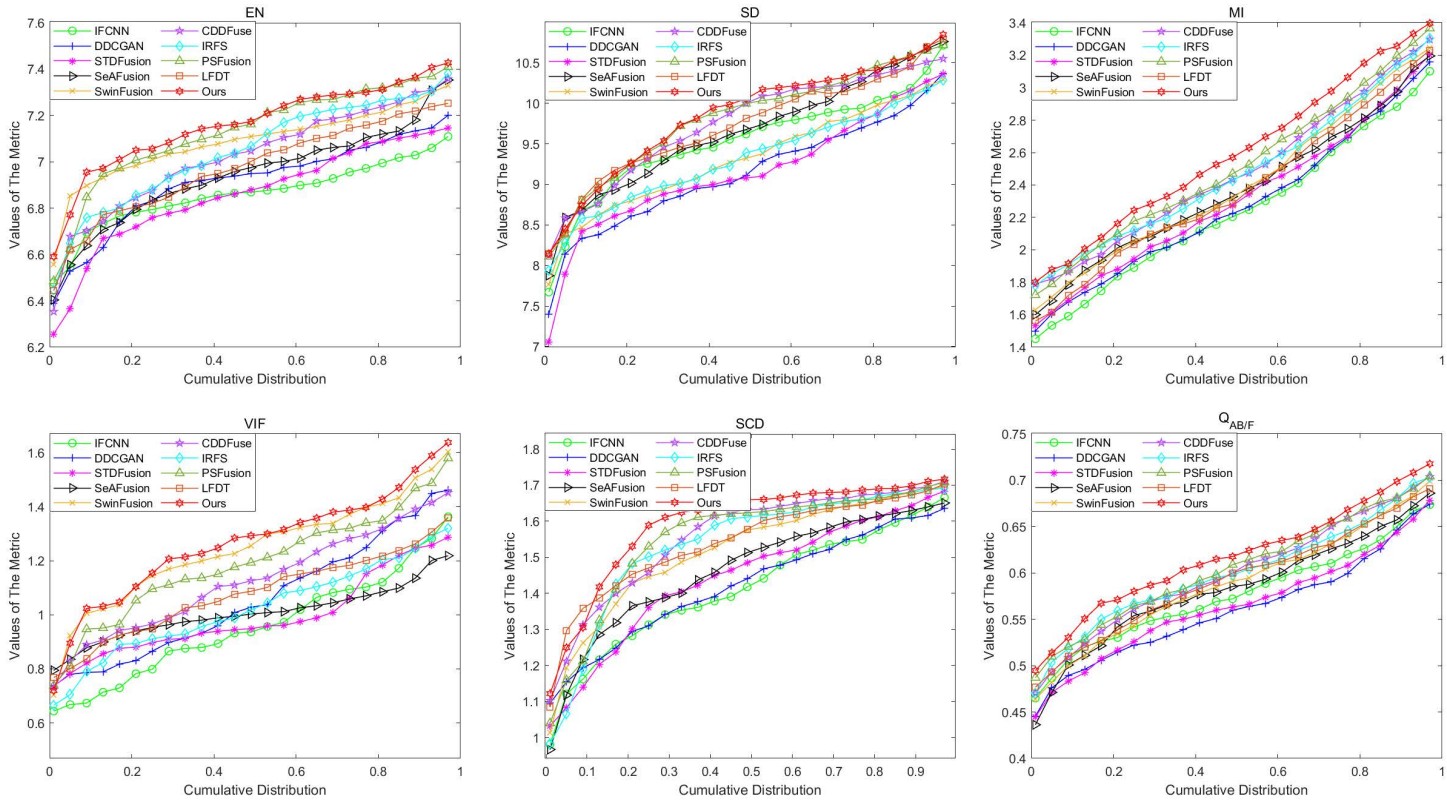

**Fig 14. Cumulative distribution of the six metrics on the M³FD dataset.**

metrics when integrated with our proposed method. These experimental results substantiate that our approach surpasses existing fusion methods in both qualitative visual perception and quantitative IoU measurements. This exceptional performance can be fundamentally attributed to two key factors: the innovative architectural design of the network framework, which optimizes feature representation, and the meticulously configured detail sub-network, which not only enhances fusion capabilities but also effectively preserves and integrates crucial semantic information throughout the processing pipeline.

## 4.4. Ablation experiments

Extensive research on deep learning-based fusion frameworks has established that the fusion performance is fundamentally determined by the intrinsic architectural design of the framework. To systematically investigate the impact of our framework's structural components, we conducted a comprehensive ablation study. The proposed framework introduces two innovative architectural features: an interspersed LT mechanism within the backbone network, which leverages Transformer architecture for multi-attention feature extraction, and a dedicated detail information extraction module that operates synergistically with the hierarchical feature extraction process in the main network, complemented by an INN-based detail integration strategy. To quantitatively assess the individual contributions of these components, we implemented three controlled experimental configurations: removal of the detail features extraction module (-w/o DFEM), elimination of the LT mechanism (-w/o LT) in the backbone network, and exclusion of the INN component (-w/o INN) in the detail integration phase. The comparative experimental results, as illustrated in Fig 16, provide valuable insights into the performance impact of each architectural component.

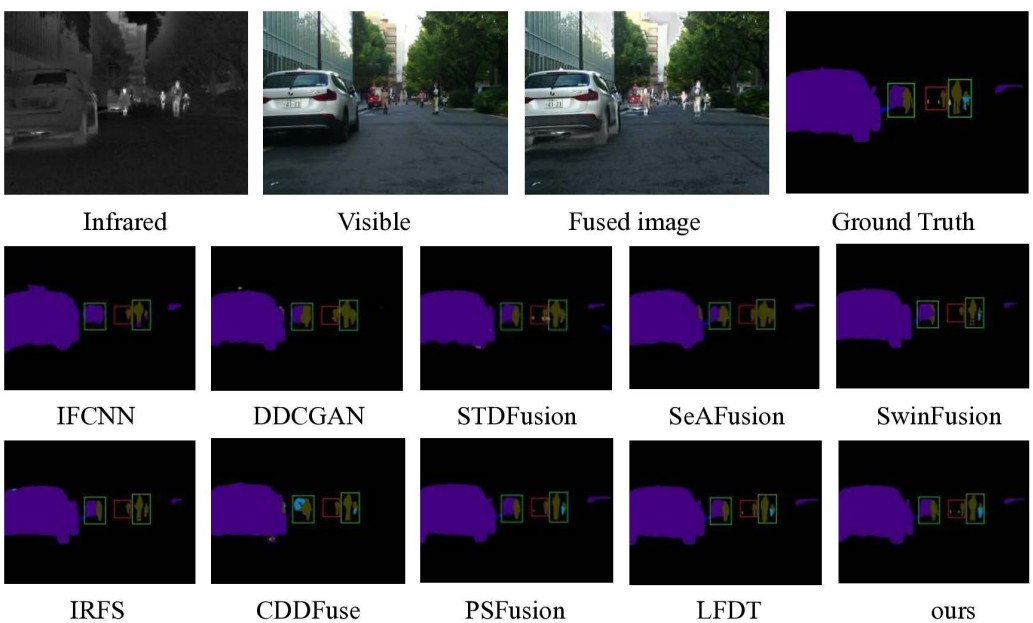

**Fig 15. Segmentation results of ten fusion methods under BANet.**

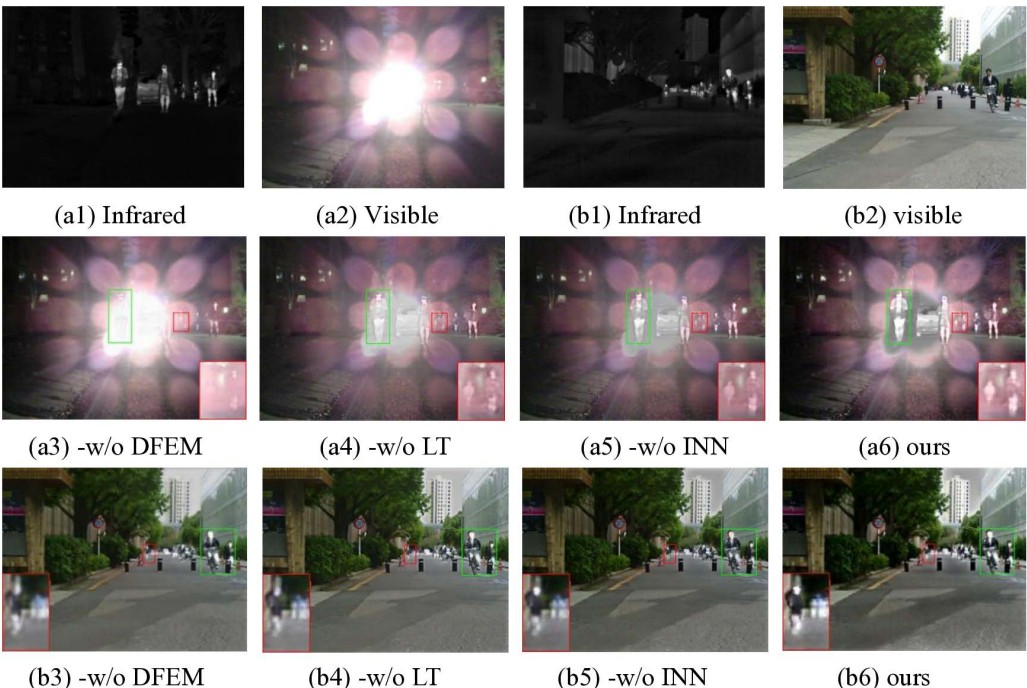

**Fig 16. Visualization results of ablation experiments.**

**Table 3. Segmentation of MFNet data under three segmentation models. Where STDFusion, SeAFusion, SwinFusion, CDDFuse and PSFusion are abbreviated as STDF., SeAF., SwinF., CDDF. and PSF. The highest value is marked in bold.**

| | | Infrared | Visible | IFCNN | DDCGAN | STDF. | SeAF. | SwinF. | IRFS | CDDF. | PSF. | LFDT | ours |
|---|---|---|---|---|---|---|---|---|---|---|---|---|---|
| Background | BANet | 98.24 | 98.26 | 98.42 | 98.48 | 98.54 | 98.60 | 98.49 | 98.42 | 98.55 | 98.69 | 98.70 | 98.72 |
| | SegFormer | 98.35 | 98.41 | 98.50 | 98.52 | 98.61 | 98.66 | 98.64 | 98.59 | 98.73 | 98.72 | 98.68 | 98.85 |
| | SegNeXt | 98.44 | 98.47 | 98.59 | 98.63 | 98.52 | 98.70 | 98.66 | 98.62 | 98.75 | 98.73 | 98.65 | 98.80 |
| Car | BANet | 87.33 | 89.03 | 90.01 | 89.88 | 90.24 | 90.38 | 90.20 | 90.25 | 91.12 | 91.66 | 91.40 | 91.44 |
| | SegFormer | 88.85 | 90.76 | 90.23 | 89.44 | 90.88 | 91.01 | 91.05 | 89.87 | 91.24 | 91.55 | 91.62 | 91.83 |
| | SegNeXt | 89.97 | 91.18 | 90.89 | 91.61 | 91.56 | 91.81 | 91.89 | 91.76 | 91.88 | 91.81 | 91.78 | 91.98 |
| Person | BANet | 70.46 | 59.94 | 71.56 | 72.90 | 72.82 | 74.56 | 72.24 | 73.36 | 75.56 | 76.81 | 76.85 | 76.95 |
| | SegFormer | 72.28 | 65.54 | 74.24 | 74.10 | 74.56 | 75.14 | 75.11 | 75.23 | 75.83 | 76.43 | 76.40 | 76.56 |
| | SegNeXt | 72.47 | 65.76 | 75.87 | 75.68 | 74.65 | 75.93 | 75.32 | 75.96 | 76.56 | 77.26 | 77.20 | 77.55 |
| Bike | BANet | 69.23 | 70.00 | 72.21 | 71.96 | 72.26 | 72.09 | 70.31 | 71.76 | 72.68 | 72.93 | 72.88 | 73.05 |
| | SegFormer | 70.28 | 71.48 | 72.42 | 72.26 | 72.84 | 72.47 | 72.30 | 72.97 | 73.25 | 73.07 | 73.00 | 73.34 |
| | SegNeXt | 70.75 | 72.33 | 73.24 | 73.30 | 73.43 | 72.12 | 72.95 | 73.22 | 73.52 | 73.64 | 73.58 | 73.88 |
| Curve | BANet | 58.74 | 60.69 | 62.58 | 62.46 | 63.85 | 63.26 | 62.53 | 63.88 | 64.21 | 64.48 | 64.21 | 64.23 |
| | SegFormer | 59.15 | 59.70 | 62.74 | 62.98 | 63.86 | 64.50 | 62.32 | 63.96 | 64.26 | 63.76 | 63.98 | 64.66 |
| | SegNeXt | 60.49 | 61.42 | 63.54 | 63.83 | 64.23 | 64.48 | 62.55 | 63.35 | 64.52 | 64.76 | 64.72 | 64.94 |
| Car Stop | BANet | 68.85 | 71.43 | 72.03 | 71.66 | 72.63 | 74.38 | 75.85 | 73.64 | 74.68 | 74.32 | 74.80 | 74.93 |
| | SegFormer | 70.02 | 77.49 | 79.28 | 78.88 | 79.42 | 79.12 | 77.56 | 78.75 | 79.53 | 80.01 | 80.12 | 80.21 |
| | SegNeXt | 74.93 | 78.24 | 79.53 | 79.31 | 79.94 | 79.91 | 78.23 | 79.62 | 80.34 | 80.15 | 80.24 | 80.69 |
| Guardrail | BANet | 65.57 | 77.90 | 80.36 | 80.12 | 82.13 | 84.34 | 85.06 | 84.73 | 85.34 | 85.60 | 85.52 | 85.87 |
| | SegFormer | 65.51 | 83.52 | 84.97 | 82.22 | 83.53 | 85.04 | 83.06 | 84.97 | 85.37 | 86.82 | 86.05 | 86.31 |
| | SegNeXt | 76.48 | 81.26 | 82.55 | 82.45 | 82.73 | 81.48 | 83.40 | 82.99 | 84.57 | 85.92 | 85.88 | 86.04 |
| Color cone | BANet | 56.93 | 63.42 | 64.82 | 64.44 | 65.28 | 66.21 | 63.88 | 63.97 | 65.29 | 66.47 | 66.37 | 66.76 |
| | SegFormer | 56.27 | 65.27 | 64.39 | 63.67 | 64.73 | 65.63 | 63.14 | 64.62 | 65.77 | 68.21 | 67.58 | 68.04 |
| | SegNeXt | 59.71 | 62.40 | 64.39 | 64.26 | 65.24 | 67.75 | 65.74 | 66.53 | 68.13 | 67.17 | 67.08 | 67.82 |
| Bump | BANet | 72.72 | 75.31 | 76.59 | 75.56 | 76.48 | 77.12 | 80.36 | 80.05 | 81.22 | 81.01 | 81.22 | 81.52 |
| | SegFormer | 74.70 | 78.38 | 79.02 | 78.64 | 79.57 | 76.39 | 81.12 | 80.36 | 80.81 | 80.28 | 80.18 | 81.23 |
| | SegNeXt | 74.26 | 80.00 | 78.18 | 76.52 | 77.62 | 80.39 | 76.72 | 77.93 | 78.91 | 79.58 | 79.76 | 79.89 |
| mIoU | BANet | 72.01 | 74.00 | 76.83 | 76.41 | 77.15 | 76.35 | 76.99 | 77.43 | 78.28 | 78.77 | 78.60 | 78.91 |
| | SegFormer | 72.82 | 76.73 | 77.72 | 77.36 | 78.02 | 78.38 | 77.70 | 78.12 | 78.82 | 79.63 | 79.74 | 79.83 |
| | SegNeXt | 75.28 | 76.79 | 78.33 | 78.26 | 78.52 | 78.78 | 78.38 | 79.26 | 79.81 | 80.38 | 80.48 | 80.53 |

The visualized comparative results in Fig 16 reveal significant performance variations across different architectural configurations. The network architecture lacking the DFEM demonstrates limited capability, primarily extracting only generic features from source images. This limitation is particularly pronounced in the first experimental set, where the importance of the detail module becomes crucial under challenging environmental conditions. A representative example can be observed in region Fig 16 (a3), where the network fails to effectively process overexposed infrared images in its fusion output. The architecture without the LT component produces visually flat fusion results, characterized by insufficient feature representation and sub-optimal brightness adjustment, particularly in target regions. While the network configuration without the INN module shows relatively improved performance compared to the previous two cases, it still under-performs in critical aspects, particularly in the accurate representation of pedestrians and distant targets, when compared to our complete proposed method. Furthermore, by comparing the quantitative metrics in Table 4, we can be more confident that the key components of our fusion framework make unique and significant contributions to the overall performance, holding important practical implications for real-world applications.

**Table 4. Average quantitative metrics of ablation experiments on the MSRS dataset.**

|          | EN            | SD            | MI            | VIF           | SCD           | $Q_{AB/F}$    |
|----------|---------------|---------------|---------------|---------------|---------------|---------------|
| -w/o DFEM | $5.542 \pm 0.562$ | $7.523 \pm 2.547$ | $2.042 \pm 1.457$ | $0.826 \pm 0.347$ | $1.432 \pm 0.245$ | $0.425 \pm 0.134$ |
| -w/o LT   | $5.723 \pm 0.452$ | $7.718 \pm 3.245$ | $2.247 \pm 1.246$ | $0.835 \pm 0.247$ | $1.503 \pm 0.256$ | $0.527 \pm 0.124$ |
| -w/o INN  | $5.985 \pm 0.824$ | $8.042 \pm 2.249$ | $2.346 \pm 1.471$ | $0.934 \pm 0.453$ | $1.632 \pm 0.204$ | $0.624 \pm 0.143$ |
| Ours      | $6.799 \pm 0.702$ | $8.420 \pm 3.104$ | $2.650 \pm 1.349$ | $1.066 \pm 0.362$ | $1.750 \pm 0.198$ | $0.677 \pm 0.171$ |

## 5. Conclusion

Building upon state-of-the-art (SOTA) fusion frameworks, we propose a novel, more robust, and higher-performance fusion network architecture. This comprehensive design integrates four key components: (1) a dual-stream backbone network combining LT and ResNet architectures, (2) a sophisticated detail extraction and integration network based on INN, (3) a basic information integration sub-network, and (4) a hierarchical semantic information extraction module. The backbone network utilizes a multi-head attention mechanism to facilitate progressive, layer-wise feature extraction, enabling comprehensive information mining from shallow to deep levels. Unlike existing CCDFuse and PSFusion methods, which neglect the extraction of intrinsic deep-level image details, our approach introduces a critical innovation: a detail-oriented sub-network. This sub-network employs deep hierarchical processing to meticulously capture textures and intrinsic features. Departing from conventional approaches that directly feed extracted information into fusion strategies, we implement a novel bottom-up progressive integration paradigm. Crucially, our architecture achieves semantic information integration through inter-layer detail extraction, thereby eliminating the need for additional network components. This design innovation substantially reduces computational complexity and parameter count. To optimize network performance, we have developed specialized correlation loss functions that jointly constrain the network's coordination during both base and detail information extraction across layers. Leveraging advanced GPU computational capabilities, we conducted extensive training and fine-tuning of the network framework. Experimental results demonstrate that our network surpasses numerous popular architectures in both qualitative visual perception and quantitative metrics. Particularly noteworthy is its exceptional performance in detail extraction under challenging environmental conditions, showcasing remarkable de-fogging capabilities and information integration efficiency. The network also excels in image clarity and color reproduction. While the network demonstrates superior performance in most aspects, we have identified areas for improvement, particularly in color rendering of specific regions such as sky areas, which occasionally appear darker than desired. Addressing this limitation will be a focus of our future research efforts, along with further optimization of the network's computational efficiency and generalization capabilities across diverse environmental conditions.

## Supporting information

**S1 File. All original code and part of the data analyzed in this study are provided as supplementary materials in a file named "code.rar".**
(RAR)

## Acknowledgments

The authors would like to extend their heartfelt appreciation to the editorial board and anonymous reviewers for their meticulous evaluation, valuable insights, and constructive recommendations, which have significantly enhanced the quality of this work.

## Author contributions

**Conceptualization:** Wei Liu.

**Formal analysis:** Fang Zhu.

**Funding acquisition:** Wei Liu.

**Investigation:** Wei Liu.

**Methodology:** Fang Zhu, Wei Liu.

**Software:** Fang Zhu.

**Writing – review & editing:** Wei Liu.

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
