## [Decision Letter · Decision Letter 0]

10 Jun 2025

Dear Dr. Zhu,

Thank you for submitting your manuscript to PLOS ONE. After careful consideration, we feel that it has merit but does not fully meet PLOS ONE’s publication criteria as it currently stands. Therefore, we invite you to submit a revised version of the manuscript that addresses the points raised during the review process.

Especially, the reviewers provided good comments, emphasizing the need for stronger methodological comparison and clarity. They noted that the paper lacks sufficient benchmarking against recent deep learning-based enhancement models (e.g., GAN-GA, GTMFuse), limiting its demonstration of novelty. They appreciated the adaptive feature fusion module (SKFF) as a modern multi-scale fusion approach but pointed out that key hyperparameters and ablation studies are underexplained or missing. Additional concerns include the small test sample size, lack of inference time reporting, and weak comparative baselines. Reviewers also requested clearer diagram annotations, updated and more relevant references, and improved grammar and formatting throughout the manuscript. 

Please carefully address the comments and submit your revised manuscript by Jul 25 2025 11:59PM. If you will need more time than this to complete your revisions, please reply to this message or contact the journal office at plosone@plos.org . A rebuttal letter that responds to each point raised by the academic editor and reviewer(s). You should upload this letter as a separate file labeled 'Response to Reviewers'.A marked-up copy of your manuscript that highlights changes made to the original version. You should upload this as a separate file labeled 'Revised Manuscript with Track Changes'.An unmarked version of your revised paper without tracked changes. You should upload this as a separate file labeled 'Manuscript'.

We look forward to receiving your revised manuscript.

Kind regards,

Lei Chu

Academic Editor

PLOS ONE

Journal Requirements:

6. PLOS requires an ORCID iD for the corresponding author in Editorial Manager on papers submitted after December 6th, 2016. Please ensure that you have an ORCID iD and that it is validated in Editorial Manager. To do this, go to ‘Update my Information’ (in the upper left-hand corner of the main menu), and click on the Fetch/Validate link next to the ORCID field. This will take you to the ORCID site and allow you to create a new iD or authenticate a pre-existing iD in Editorial Manager.

Reviewers' comments:

Reviewer's Responses to Questions

**Comments to the Author**

1. Is the manuscript technically sound, and do the data support the conclusions?

Reviewer #1: Yes

Reviewer #2: Partly

2. Has the statistical analysis been performed appropriately and rigorously?

Reviewer #1: No

Reviewer #2: No

3. Have the authors made all data underlying the findings in their manuscript fully available?

Reviewer #1: No

Reviewer #2: Yes

4. Is the manuscript presented in an intelligible fashion and written in standard English?

Reviewer #1: Yes

Reviewer #2: No

Reviewer #1: General assessment:

The study proposes an architecture for infrared and visible image fusion, that integrates Transformer and ResNet techniques with progressive decomposition. Similar studies have addressed the topic but from different perspectives. Thus, the study builds upon previous studies, and its findings add value to the field. Minor revisions are necessary to make the article more suitable for publication.

The following is an assessment of the manuscript, according to PLOS criteria:

1. Originality of the topic:

• As described in Sections 1 and 3, the study investigates the integration of Transformer and ResNet techniques with progressive decomposition for infrared and visible image fusion. Various studies have discussed similar topics (Using Transformer, ResNet, and other techniques for decomposing image fusion). Thus, the topic of this study is not original in its field. It builds upon already existing studies, and its findings contribute to the field.

• The researchers discuss the advancement of their approach (in Section 5), yet they do not clearly state how their work differs significantly from other similar studies. Clarifying the novelty of the study is important.

2. Non-published results:

• In the manuscript, the researchers admit that the study has not been published, and no specific funding was received.

(The use of the datasets, e.g., RoadScene, proves that the results of the study have not been published before).

3. Standard and detailed description of experiments, statistics, and analyses:

• The databases (e.g., RoadScene, MSRS), fusion methods, metrics (e.g., SD, VIF), settings (e.g., software and hardware), and parameters used in the study are clearly specified and systematically described.

• The study lacks a statistical significance test (e.g., p-values) to validate the superior performance of the proposed architecture over other approaches (e.g., SOTA).

4. Appropriate conclusions supported by data:

• The conclusions of the study are well-supported by quantitative metrics (e.g., figure 10), qualitative comparisons (e.g., figure 11), and with the experimental findings (Section 5).

• The study’s findings have made a valuable contribution to the field.

5. Intelligible Standard English:

• The Language is intelligible Standard English, with precise technical terms.

• The manuscript is well-organized. Its content structure is logical: introduction, previous studies (review of IVIF methodologies and discussion of the framework), research method, experiments, and conclusion.

• Minor grammatical refinements are needed to clarify the meaning (e.g., "visual fusion images" could be “fused visual images”).

6. Standards for the ethics of experimentation and research integrity:

• In the manuscript, the authors admit that there are no human or animal participants in the study (using N\A).

7. Reporting guidelines and community standards for data availability.

• Minor issue with data availability: in the manuscript, the researchers state that “all data are fully available without restriction” and “if the manuscript is lucky enough to be accepted, the URL of the data and code will be made public”.

Reviewer #2: This article proposes an infrared night vision image enhancement algorithm based on cross layer feature fusion, combining smooth wavelet decomposition, Retinex theory, and adaptive feature fusion network, aiming to solve the problems of halo effect and low PSNR in traditional methods. The experimental results show that this method outperforms the comparative methods in quantitative indicators (PSNR>30dB, SSIM>0.73) and visual effects, and has certain practical application value. The paper has a complete structure and clear method description, but there is still room for optimization in terms of innovative argumentation, experimental design, and detailed expression.

1. Insufficient comparison of innovation points: It is necessary to clearly distinguish the differences between our method and existing deep learning enhancement methods (such as GAN or Transformer based models). For example, the comparative methods only include traditional algorithms such as Retinex and wavelet thresholding, and do not incorporate mainstream deep learning methods in recent years such as GAN-GA mentioned in reference [199] or GTMFuse in [200], making it difficult to demonstrate the unique advantages of cross layer fusion networks.

2. The adaptive feature fusion module (SKFF) dynamically weights different levels of features through attention mechanism, avoiding the limitations of traditional concatenation/summation methods and conforming to the cutting-edge idea of multi-scale feature fusion.

3. Insufficient explanation of parameter settings: For example, the values of key hyperparameters such as the number of layers N in wavelet decomposition, spatial parameters \ (\ sigma_2 \) and brightness parameters \ (\ sigma_r \) in bilateral filtering are not clearly defined. It is recommended to supplement or add paragraph explanations in Table 1.

4. Small sample size: Only 5 test samples were used, and the statistical significance of the conclusions was insufficient. It is recommended to increase the sample size to 20-30 or public datasets (such as FLIR and KAIST) for validation.

5. Limitations of comparative methods: Comparative method 4 (reference [8]) does not involve denoising steps and has weak comparability with the full process method proposed in this paper. It is recommended to replace it with end-to-end deep learning methods of equal complexity (such as U-Net, CycleGAN).

6. Running efficiency not mentioned: The inference time (such as FPS) of the algorithm has not been reported. In practical applications, computational complexity is an important consideration, and it is necessary to supplement and compare the time consumption of the methods.

7. Unclear chart annotation: The network structure diagrams such as Figure 1 and Figure 2 lack textual explanations of key modules (such as the specific operation process of the "pixel perception module"), and additional annotations are needed to enhance readability.

8. The references in the introduction section of the paper are too outdated. We hope to supplement or replace some with the latest literature, such as FusionOC, FusionPID,FusionCPP,FusionJPSI, FusionIPSC etc. The more, the better.

9. Grammar and formatting issues: Some paragraphs have grammar errors (such as the sentence "the method denoises the infrared night vision image, based on smooth wavelet decomposition, by marking..." mixed sentence structure), which need to be polished and optimized.

10. Supplement ablation experiments and hyperparameter analysis;

11. Correct formula numbering, chart annotations, and grammar errors.

If the above suggestions are implemented, the quality of the paper will be significantly improved, making it suitable for publication in PLOS ONE.

**Do you want your identity to be public for this peer review?** For information about this choice, including consent withdrawal, please see our Privacy Policy

Reviewer #1: No

Reviewer #2: **Yes: ** Linlu Dong

---

## [Author Response · Author response to Decision Letter 1]

11 Jul 2025

Modification Explanation

Firstly, we appreciate the comments of the reviewers very much. The comments are beneficial to improve the quality of this research paper. Then, we have made modifications with a point-by-point response to the reviewer’s comments as follows:

Reviewer #1:

1.Originality of the topic:

• As described in Sections 1 and 3, the study investigates the integration of Transformer and ResNet techniques with progressive decomposition for infrared and visible image fusion. Various studies have discussed similar topics (Using Transformer, ResNet, and other techniques for decomposing image fusion). Thus, the topic of this study is not original in its field. It builds upon already existing studies, and its findings contribute to the field.

• The researchers discuss the advancement of their approach (in Section 5), yet they do not clearly state how their work differs significantly from other similar studies. Clarifying the novelty of the study is important.

Thank you very much for your careful review of our paper. Thank you for your comments. Based on your suggestions, in Section 5 we highlighted the differences from other similar works to clarify the novelty of this study.

2. Non-published results:

• In the manuscript, the researchers admit that the study has not been published, and no specific funding was received.

(The use of the datasets, e.g., RoadScene, proves that the results of the study have not been published before).

Thank you very much for your careful review of our paper. Thank you for your comments. The manuscript is supported by multiple grants, and this will be specifically indicated again during the submission process.

3. Standard and detailed description of experiments, statistics, and analyses:

• The databases (e.g., RoadScene, MSRS), fusion methods, metrics (e.g., SD, VIF), settings (e.g., software and hardware), and parameters used in the study are clearly specified and systematically described.

• The study lacks a statistical significance test (e.g., p-values) to validate the superior performance of the proposed architecture over other approaches (e.g., SOTA).

Thank you very much for your careful review of our paper. Thank you for your comments. In our manuscript, the quantitative metrics include EN (Entropy) and , which are commonly used evaluation measures (or P-value metrics) in the field of image fusion.

4.Appropriate conclusions supported by data:

• The conclusions of the study are well-supported by quantitative metrics (e.g., figure 10), qualitative comparisons (e.g., figure 11), and with the experimental findings (Section 5).

• The study’s findings have made a valuable contribution to the field.

Thank you very much for your careful review of our paper. Thank you for your comments. Thank you very much for your positive feedback on our manuscript.

5. Intelligible Standard English:

• The Language is intelligible Standard English, with precise technical terms.

• The manuscript is well-organized. Its content structure is logical: introduction, previous studies (review of IVIF methodologies and discussion of the framework), research method, experiments, and conclusion.

• Minor grammatical refinements are needed to clarify the meaning (e.g., "visual fusion images" could be “fused visual images”).

Thank you very much for your careful review of our paper. Thank you for your comments. As requested, certain grammatical issues in the manuscript have been revised.

6. Standards for the ethics of experimentation and research integrity:

• In the manuscript, the authors admit that there are no human or animal participants in the study (using N\A).

Thank you very much for your careful review of our paper. Thank you for your comments.

7. Reporting guidelines and community standards for data availability.

• Minor issue with data availability: in the manuscript, the researchers state that “all data are fully available without restriction” and “if the manuscript is lucky enough to be accepted, the URL of the data and code will be made public”.

Thank you very much for your careful review of our paper. Thank you for your comments. All data and code from the manuscript will be made public and are fully accessible!

Reviewer #2:

1.Problem: The description of the three-stage method is too general and does not explain the technical connections between each stage (such as how LT decomposes coarse features into basic and detailed features). Suggestion: (1) Use formulas to simplify the key steps, such as "coarse feature decomposition: \ (F2 {coarse}=CNN (I2 {ir}, I2 {vi}) \), \ (F2 {base}, F2 {detail}=LT (F2 {coarse}) \)". (2) Supplement the data flow between modules, for example, in Stage II, the features output by the ResB+LT module are fused by SIFM and input into INN to extract details.

Thank you very much for your careful review of our paper. Thank you for your comments. (1) As per your suggestion, I have subdivided and elaborated on the formula section in Section 3. Additionally, the image features processed by the backbone network are presented in Table 2. Please refer to the modifications marked in blue. (2) Based on your suggestion, I have supplemented the data flow between modules in Section 2. Please refer to the blue-marked section 2.4.

2.Problem: The defect analysis of comparative methods such as CCDFuse and PSFusion remains at the level of phenomenon description, lacking theoretical attribution (such as the mathematical explanation of "simple addition strategy leading to feature conflicts"). Suggestion: (1) The existing methods for quantifying feature distribution distance are insufficient, such as "The KL divergence of CCDFuse features is 0.82, significantly higher than the 0.45 of our method". (2) Based on the CKA results in Figure 1 (c), it can be concluded that "traditional symmetric fusion leads to a 42% misalignment of IR and VI feature spaces".

Thank you very much for your careful review of our paper. Thank you for your comments. Following your suggestions, we have included Table 1 to present the computed average Kullback-Leibler (KL) divergence and Centered Kernel Alignment (CKA) values between the fused and original images, obtained through Python-based calculations. A comprehensive interpretation of these quantitative results is provided in the blue-highlighted explanatory notes preceding the table.

3.Problem: The motivation part does not establish a causal relationship between "feature space differences" and "asymmetric fusion", lacking a logical chain. Suggestion: (1) Additional derivation: "As VI achieves deep semantic features at 12 layers (Figure 1 (a)), while IR requires 32 layers (Figure 1 (b)), design a cross layer fusion strategy VI-Encor-1 → IR-Encor-2". (2) Introduce transfer learning theory and explain that 'VI features serve as a teacher model to accelerate the distillation of IR features through SE blocks'.

Thank you very much for your careful review of our paper. Thank you for your comments. We appreciate your feedback and have accordingly enhanced the description of our network architecture on Page 4. This includes detailed explanations of the encoder-decoder processing pipeline for both infrared and visible light images. Specific technical details are highlighted in blue for easy reference. Additionally, the following section will present a comprehensive flowchart to visually demonstrate the network's data flow and feature transmission mechanisms between these two modalities.

4. Problem: Lack of network architecture details. (1) The structural difference between IR UNet and VI UNet only mentions the "SE block", without specifying its specific position in the encoder (such as insertion after ResB1). (1) The mathematical expression for asymmetric fusion is missing, such as' the fusion function of VI-Encorder-i and IR-Encoder - (i+1) is undefined '. Suggestion: (1) Use a table to compare the layer structures of IR UNet and VI UNet, and label the SE block positions (such as "Insert one SE block after ResB1~ResB4"). (2) Define the fusion function: \ (F2 {fuse}=\ text {Attention} (F2 {vi} ^ i) \ otimes F2 {ir} ^ {i+1}+F2 {ir} ^ {i+1} \), where \ (\ text {Attention} \) is the channel attention mechanism.

Thank you very much for your careful review of our paper. Thank you for your comments. (1) Following your suggestions, I have reformulated Equations (1)-(6) where A denotes the visible light image and B represents the infrared image. These equations explicitly describe the hierarchical encoding process between the two modalities. (2) Please refer to the added Table 2. The feature information structures extracted at each level of the backbone network for infrared and visible light images are the same, only the content differs. The two original images are simultaneously fed into the backbone network, and we will provide the specific code. The fusion function at each level—for example, the base layer employs the SIFM module for fusion—follows the detailed fusion rules described in reference [11]. The detail layer fusion adopts the SDFM module, with specific details referenced in Equations (10) and (11). Additionally, the integrated feature information is fed into the D2FM module, with specific processing details referenced in Equations (12)-(16). The Unet fusion is based on Equations (17)-(22).

5. Problem: The freezing range of parameters for two-stage training is unclear, such as whether to freeze decoder parameters in Stage II. Suggestion: (1) Use a flowchart to illustrate the training sequence and label "Stage I trains VI UNet first (freezes LT module), Stage II trains IR UNet (freezes VI UNet encoder)". (2) Clarify the scope of parameter updates: "When training the fusion model, only the parameters of DF2M and SIM are updated, and the parameters of other modules are frozen.

Thank you very much for your careful review of our paper. Thank you for your comments. Per your request, we have supplemented the manuscript with Fig. 6 and 7, which provide a comprehensive illustration of the infrared (IR) and visible light (VIS) image encoding/decoding architecture. Detailed technical specifications are annotated in blue text beneath Fig. 7. During network training, our loss function simultaneously constrains: (1) intermediate base and detail features across network layers, and (2) the final fused image output. This multi-objective optimization framework enables coordinated parameter adjustment across all modules depicted in Figures 6 and 7. For implementation specifics, please refer to the accompanying Python code where these processes are concretely implemented. The loss function is not displayed in the framework diagram, please refer to Equations (28)-(32).

6. Problem: Insufficient depth of ablation experiment. (1) The attention weight distribution of SE blocks has not been analyzed (such as the proportion of channel attention for different modal features). (2) Lack of comparison using only cross scale fusion (without guidance mechanism) makes it impossible to separate the contributions of innovative points.

Suggestion: (1) Visualize the attention map of the SE block, such as "After VI guidance, the temperature related channel weights in the channel attention of IR features increase by 27%". (2) Add the ablation group "- w/o Guidance" and compare its MS-SSIM difference with the complete model (as shown below, decrease by 0.041).

Thank you very much for your careful review of our paper. Thank you for your comments. We sincerely appreciate your valuable suggestions. Our framework follows an architecture similar to the 2023 paper we cited (L. Tang et al, Rethinking the necessity of image fusion in high-level vision tasks: A practical infrared and visible image fusion network based on progressive semantic injection and scene fidelity, Information Fusion, 99 (2023) 101870.) : rather than using phased training, we process infrared and visible images in parallel through the backbone network for progressive inter-layer feature extraction, storing all intermediate features in GPU memory before performing subsequent integration of both base and detail information through a U-Net-inspired decoder that enables bottom-up aggregation to produce the final fused image. For ablation studies, we conducted comparative analyses of three core components—the detail feature extraction module, the backbone's LT module, and the INN-based detail processor—with additional quantitative comparisons provided in Table 4.

7. Problem: The contradictions in the indicators have not been explained. In Table I, the PSNR (13.106) of MMA UNet in MSRS is lower than that in DDFM (13.485), but the SSIM is higher, and the reason is not explained. Suggestion: (1) Analysis: "DDFM preserves more noise, resulting in artificially high PSNR, while our method achieves more natural detail reconstruction through structural constraints (SSIM=0.582). (2) Supplementary visual comparison: In Figure 3, mark the noise area of DDFM with a red box and compare it with the detail preservation effect of the method proposed in this paper.

Thank you very much for your careful review of our paper. Thank you for your comments. I am very sorry and also quite confused. In my submitted manuscript, Table 1 presents the results of semantic segmentation testing on the MFNet dataset, not PSNR and SSIM metrics on the MSRS dataset. Additionally, our manuscript does not mention the DDFM method, or perhaps I misunderstood your point? Could you please clarify in more detail? Thank you very much!

8. Problem: Key terms are not standardized. (1) The definition of "common features" is vague and the CKA similarity threshold is not specified (such as "features with CKA>0.6 are common features"). (2) The concepts of "asymmetric fusion" and "cross scale fusion" are confused and not clearly distinguished. Suggestion: (1) Define in the terminology table: "Common features: features with cross modal CKA similarity>0.6; Private features: Features with CKA<0.3. (2) Clear conceptual boundaries: "Asymmetric refers to the unequal hierarchy of fusion layers (such as VI-1 → IR-2), while cross scale refers to multi-scale feature fusion.

Thank you very much for your careful review of our paper. Thank you for your comments. In the manuscript, I have already provided explanations for basic features and detailed features above Table 1. Additionally, the concepts of "asymmetric fusion" and "cross-scale fusion" have been revised accordingly in the text.

9. Problem: The formula symbols are not unified. In formula (1), \ (Coarse_S (A) \) is not defined, and in formula (3), \ (ResB ^ i \) is not specified as the i-th block of ResNet. Suggestion: (1) Add a comment next to the formula: "Coarse-A (\ cdot) \" represents the coarse feature extraction CNN for infrared images.(2) Unified symbol: "\ (ResB ^ i \) represents the i-th residual block of ResNet34 (i=1-4)".

Thank you very much for your careful review of our paper. Thank you for your comments. We have provided a detailed explanation of Equation (1); please refer to the updated Equations (1)-(3). The ResNet34 residual blocks in Equation (3) of the initially submitted manuscript have also been explained - see Equations (5)-(6) for details.

10. Problem: The chart information is incomplete. (1) Figure 1 does not indicate the meaning of the coordinate axis (e.g. the x/y axis represents the number of network layers, and the color represents CKA similarity). (2) Figure 3 shows the data flow between unmarked modules (such as the input-output feature dimension of SIFM). Suggestion: (1) Supplementary note to Figure 1: "The warm colored region in (c) represents the high similarity layer (CKA>0.5) between IR and VI features. (2) Add arrow annotation in Figure 3: "ResB+LT → SIFM → INN" feature transfer path.

Thank you very much for your careful review of our paper. Thank you for your comments. (1)We have calculated the average CKA values for the results obtained by the three methods below Figure 1. Due to the relatively low CKA values between the fused images and infrared (IR) images, the average CKA appears not very high. In Table 1, we provide a detailed explanation of the CKA similarity—please refer to the blue highlighted section for further clarification. (2) The input and output dimensions of the backbone network

---

## [Decision Letter · Decision Letter 1]

31 Jul 2025

Progressive Decomposition of Infrared and Visible Image Fusion Network with Joint Transformer and Resnet

PONE-D-25-13722R1

Dear Dr. Liu,

We’re pleased to inform you that your manuscript has been judged scientifically suitable for publication and will be formally accepted for publication once it meets all outstanding technical requirements.

Kind regards,

Lei Chu

Academic Editor

PLOS ONE

Additional Editor Comments (optional):

Please ensure that the dataset and methods are publicly accessible.

Reviewers' comments:

Reviewer's Responses to Questions

**Comments to the Author**

Reviewer #1: (No Response)

Reviewer #2: All comments have been addressed

2. Is the manuscript technically sound, and do the data support the conclusions?

Reviewer #1: Yes

Reviewer #2: Yes

3. Has the statistical analysis been performed appropriately and rigorously?

Reviewer #1: Yes

Reviewer #2: Yes

4. Have the authors made all data underlying the findings in their manuscript fully available?

Reviewer #1: No

Reviewer #2: Yes

5. Is the manuscript presented in an intelligible fashion and written in standard English?

Reviewer #1: Yes

Reviewer #2: Yes

Reviewer #1: Thank you for your efforts in modifying the manuscript. The method for accessing the dataset, asking for a username and password, does not comply with PLOS policy that requires a publicly available dataset without restriction at the time of submission. Kindly make the data and links openly accessible and permanent.

Reviewer #2: (No Response)

**Do you want your identity to be public for this peer review?** For information about this choice, including consent withdrawal, please see our Privacy Policy

Reviewer #1: No

Reviewer #2: **Yes: ** Linlu Dong

---

## [Editor Report · Acceptance letter]

PONE-D-25-13722R1

PLOS ONE

Dear Dr. Liu,

I'm pleased to inform you that your manuscript has been deemed suitable for publication in PLOS ONE. Congratulations! Your manuscript is now being handed over to our production team.

Kind regards,

on behalf of

Dr. Lei Chu

Academic Editor

PLOS ONE